

# Diatom responses and geochemical feedbacks to environmental changes at Lake Rauchuagytgyn (Far East Russian Arctic)

Boris K. Biskaborn[1], Amy Forster[1,2], Gregor Pfalz[1,3], Lyudmila A. Pestryakova[4], Kathleen Stoof-Leichsenring[1], Jens Strauss[5], Tim Kröger[1,6], Ulrike Herzschuh[1,2,7]

[1]Alfred Wegener Institute Helmholtz Centre for Polar and Marine Research, Polar Terrestrial Environmental Systems, Telegrafenberg A45, 14473 Potsdam, Germany
[2]University of Potsdam, Institute for Biochemistry and Biology, 14469 Potsdam, Germany
[3]University of Potsdam, Institute of Geosciences, 14469 Potsdam, Germany
[4]North-Eastern Federal University of Yakutsk, 677000 Sakha Republic, Russia
[5]Alfred Wegener Institute Helmholtz Centre for Polar and Marine Research, Permafrost Research, 14473 Potsdam, Germany
[6]Technische Universität Berlin, 10623 Berlin, Germany
[7]University of Potsdam, Institute of Environmental Science and Geography, 14469 Potsdam, Germany

*Correspondence to:* Boris K. Biskaborn (boris.biskaborn@awi.de)

**Abstract.** This study is based on multiproxy data gained from a [14]C-dated 6.5 m long sediment core and a [210]Pb-dated 23 cm short core retrieved from Lake Rauchuagytgyn in Chukotka, Arctic Russia. The main objectives are to reconstruct the environmental history and ecological development of the lake during the last 29k years and to investigate the main drivers behind bioproduction shifts. The methods comprise age-modeling and accumulation
rate estimation, light-microscope diatom species analysis (74 samples), organic carbon, nitrogen, and mercury analysis. Diatoms have appeared in the lake since 21.8 cal ka BP and are dominated by planktonic *Lindavia ocellata* and *L. cyclopuncta*. Around the Pleistocene-Holocene boundary, other taxa including planktonic *Aulacoseira* and benthic fragilarioid (*Staurosira*) and achnanthoid species increase in their abundance. There is strong correlation between variations of diatom valve accumulation rates (DAR, mean 176.1 $10^9$ valves $m^2$ $a^1$),
organic carbon accumulation rates (OCAR, mean 4.6 g $m^{-2}$ $a^{-1}$), and mercury accumulation rates (HgAR, mean 63.4 µg $m^{-2}$ $a^{-1}$). We discuss the environmental forcings behind shifts in diatom species and found responses of key-taxa to the cold glacial period, postglacial warming, Younger Dryas, and the Holocene Thermal Maximum. The short core data likely suggest recent change of the diatom community at 1907 CE related to human-induced environmental change. Significant correlation between DAR and OCAR in the Holocene interglacial indicates
within-lake bioproduction as the main source of carbon deposited in the lake sediment. During both glacial and interglacial episodes HgAR is mainly bound to organic matter in the lake associated to biochemical substrate conditions. There were only ambiguous signs of increased HgAR during the industrialization period. We conclude that pristine Arctic lake systems can serve as $CO_2$ and Hg sinks during warming climate driven by insolation-enhanced within-lake primary productivity. Maintaining intact natural lake ecosystems should therefore be of
interest to future environmental policy.

## 1 Introduction

Today, northern and mountain regions warm faster than elsewhere on Earth, putting cold freshwater systems at risk for loss of ecosystem services (Ipcc, 2021). Paleoenvironmental research, however, still lack sufficient geographical coverage in the eastern Russian Arctic (Kaufman et al., 2020; Mckay et al., 2018; Sundqvist et al.,



2014). Arctic lakes are powerful archives of past climate information because they respond rapidly to external forcing on their catchments (Biskaborn et al., 2021b; Nazarova et al., 2021; Subetto et al., 2017). Effects of both, past climate changes and modern human impacts during the industrial period, including mercury contamination of pristine ecosystems have been demonstrated for remote Siberian lake ecosystems (Biskaborn et al., 2021a). In paleolimnological research many studies are based on concentrations of fossil remains and geochemical

compounds in the sediments. Within the eastern Arctic, yet only few studies managed to accomplish reconstruction of accumulation rates of these sediment constituents, possibly owed to limited age-controls (Vyse et al., 2021).

The Pleistocene-Holocene transition from glacial to interglacial climates commonly reveals the major change within biotic and geochemical sediment components and is well detectable in sufficiently old lake sediments. Shorter and less powerful climate events are less distinctly represented in low-accumulation systems, and their

impacts on lake ecosystems in sparsely covered areas is not yet sufficiently understood (Kaufman et al., 2004; Subetto et al., 2017; Biskaborn et al., 2016; Renssen et al., 2012). One of the most known groups of photosynthetic organisms in Arctic lakes are diatoms (Smol and Stoermer, 2010). They are siliceous microalgae (Bacillariophyceae) that form opaline valves which preserve excellent in lake mud and allow identification up to highest species levels by light microscope analysis (Battarbee et al., 2001). Diatoms as a group are one of the

major primary producers in aquatic environments, contributing to the global net primary production at about 25% (Stoermer and Smol 2001). Diatom communities respond to numerous environmental forcings including hydrochemical changes, seasonal climate shifts (duration of ice cover), and physical habitat changes (Hoff et al., 2015; Douglas and Smol, 2010; Pestryakova et al., 2018; Herzschuh et al., 2013; Biskaborn et al., 2012; Biskaborn et al., 2013; Palagushkina et al., 2017). Diatom productivity has been estimated from Si/Al ratios (Vyse et al.,

2020), valve concentrations (Biskaborn et al., 2012), and biogenic opal concentrations (Meyer et al., 2022). However, there is yet only sparse information available in the literature addressing the contribution of aquatic bioproduction, i.e. diatom primary producers, to accumulation rates of organic matter over different climate stages (Biskaborn et al., 2021b).

Palaeoenvironmental records that include the last glacial maximum (LGM) are sparse in the vast region of

Chukotka (Vyse et al., 2020). Lake archives exceeding the LGM were published for Lake El'gygytgyn (Melles et al., 2007), Lake Ilirney (Vyse et al., 2020; Andreev et al., 2021), and Lake Rauchuagytgyn (Vyse et al., 2021), of which the latter is subject of our study. There is still insufficient knowledge about feedbacks of specific climate events, such as the Younger Dryas cooling (Andreev et al., 2021; Kokorowski et al., 2008) or the Holocene Thermal Maximum (Renssen et al., 2012), to lake primary producers. However, short-term and fast events that

could compare to the pace of recent climate change are of specific interest to understand today's climate-ecosystem relationships.

Over the last years influences of past and recent climate changes to diatom assemblage shifts were investigated in lakes records in Yakutia (Kostrova et al., 2021; Courtin et al., 2021; Biskaborn et al., 2021b), accompanied by lake ecosystem feedbacks and long-distance heavy metal contamination (Biskaborn et al., 2021a). This study was

accordingly set up to test whether similar palaeolimnological responses to climate and anthropogenic impacts exist in very remote areas in Chukotka. In our paper we present new diatom records and biogeochemical data based on the published chronological sediment records and climate reconstructions of lake Rauchuagytgyn (Andreev et al., 2021; Vyse et al., 2021). Our objectives on Lake Rauchuagytgyn are to (1) reconstruct accumulation of diatoms since the last glacial in comparison to bulk organic carbon accumulation, (2) to investigate the main drivers behind



assemblage and bioproduction shifts, and (3) to compare past natural and recent mercury loads to test for potential heavy metal contamination of remote pristine ecosystems.

## 2 Study Site

The catchment of Lake Rauchuagytgyn (67.82° N, 168.7° E, elevation 625 m a.s.l, surface area 6.1 km$^2$, maximum water depth 36 m, catchment area 214.5 km$^2$) is located in the north-western Anadyr Mountains of Chukotka in
the northeastern Russian Arctic (Fig. 1). The lake's main inflows are situated at the southern margin and a few outflows drain the lake to the north and the sides. Glacial activity in the catchment is preserved by moraine structures north of the lake and in surrounding glacial cirques (Glushkova, 2011; Vyse et al., 2021). The basement of the study site consists of silicic-intermediate lithology, represented by cretaceous Andesite (Zhuravlev et al., 1999). The area is characterized by strong Arctic continental climate with mean annual air temperatures of -11.8
°C, mean July and January temperatures are 13 °C and -30 °C, respectively, while annual precipitation is at ca. 200 mm (Menne et al., 2012). Open herb- and graminoid tundra, with tree-occurrence only in lower elevations and close to rivers, characterizes the surrounding landscape (Huang et al., 2020; Shevtsova et al., 2020).

## 3 Materials and Methods

### 3.1 Field work

Field work and coring activities at Lake Rauchuagytgyn (Fig. 1) were performed by helicopter expeditions in July 2016 and July 2018. We used a hand-held echo sounder and a UWITEC gravity corer (60 mm) to retrieve short core 16-KP-04-L19B core with 23 cm length in summer 2016 at a 31.0 m deep part of the lake at (N 67.7888, E 168.7380). After a few hours the core was subsampled in 0.5 to 1 cm slices before transport in dark and cool conditions. A longer parallel core from the same site and expedition was analyzed for pollen and radiocarbon-
based chronology (Andreev et al., 2021).

In summer 2018 we used an INNOMAR SES-2000 compact parametric sub-bottom profiler to locate the coring location in the southern sub-basin and retrieved a long core (EN18218, ca. 6.5 m) at 29.5 m water depth using a UWITEC Niederreiter 60 mm piston coring system operated on a platform at anchor (N 67.7894, E 168.7335). Coring, processing and sediment-geochemistry of this sediment core was already described by Vyse et al. (2021).

### 3.2 Chronology

For the chronology of long core EN18218, we used the LANDO age-depth modelling approach  In the current version (v1.3), LANDO combines the output of five age-depth modelling software (Bacon, Bchron, clam, hamstr, Undatable) in a single interactive computing platform described in Pfalz et al. (2022). The advantage of this approach over a single age-depth model is that the combined model takes multiple age-depth uncertainty ranges
into consideration and reduces biases towards overinterpretation. We have updated the published sedimentation rate (SR) values for EN18218 accordingly, based on the same 23 radiocarbon dates in Vyse et al. (2021) shown in Table 1. According to Vyse et al. (2021) we added the same age offset to the data derived from the surface sample (785 ± 31 a BP), which corresponds to 853 ± 31 years when the 2018 CE expedition year is taken into account.



To date the common era over the industrial period in short core 16-KP-04-L19B, freeze-dried sub-samples were analyzed for [210]Pb and [137]Cs activities by direct gamma assay in the Liverpool University Environmental Radioactivity Laboratory, using Ortec HPGe GWL series well-type coaxial low background intrinsic germanium detectors (Appleby et al., 1986). [210]Pb and [137]Cs were measured from its gamma emissions at 46.5 keV and at 662 keV, respectively. Accuracies of used detectors were determined using calibrated standard sources of known activity. The effect of self-absorption of low-energy gamma rays was used for corrections (Appleby et al., 1992).

To model the chronology of the short core, we used the R package "rPlum" (Blaauw et al., 2021) to apply a Bayesian framework to determine the chronology based on [210]Pb measurements (Hunter et al., 2022; Aquino-López et al., 2018). We used 18 measurements of supported and unsupported [210]Pb (Table 2) within Plum, while assuming a varying supply of unsupported [210]Pb for the model (Figure 2a.2b). We corrected the Plum model by constraining it with the [137]Cs peak between 3 and 3.5 cm (Figure 2c), which is attributed to the high point in atomic

weapons testing in 1963 (Appleby, 2001; Hunter et al., 2022).

### 3.3 Biogeochemistry and mercury analysis

To gain information about the productivity in the lake we analyzed total organic carbon (TOC), total carbon (TC) and total nitrogen (TN) from 20 short core samples 16-KP-04-L19B N from 66 samples from the long core samples EN18218. 25 samples below 220 cm in EN18218, however, revealed TN values below the detection limit (0.1

wt%) and are therefore not displayed. TOC and total inorganic carbon (TIC) were detected using a Vario SoliTOC cube elemental analyser (Elementar Analysensysteme GmbH) following combustion at 400 °C for organic carbon, and 900 °C for TIC. The sum of TOC and TIC was used to estimate TC. TN was measured using a rapid MAX N exceed (Elementar Analysensysteme GmbH). The data was used to calculate the TOC/TN ratio, using a factor of 1.167, which is the ratio of the atomic weights of nitrogen (14.007 amu, atomic mass unit) and carbon (12.001

amu) to obtain the atomic ratio TOC/TN$_{atomic}$ following Meyers and Teranes (2002). TOC from EN18218 was used from Vyse et al. (2021) to estimate TOC/TN$_{atomic}$ ratios for the long core.

Total mercury (THg) was analyzed in 20 samples from core 16-KP-04-L19B and 32 samples from core EN18218. We determined the THg in solid material by thermal decomposition, amalgamation and atomic absorption spectrophotometry using a Direct Mercury Analyzer (DMA-80 evo; MLS-MWS GmbH). The solid samples were

weight into 1 ml metal boats which are then combusted at about 750°C under a flow of oxygen, and the Hg in the off-gases is trapped as amalgam on a gold sieve. In a subsequent step, Hg is released and its amount is determined by atomic absorption spectroscopy. We used the certified reference material BCR® - 142R (67µg kg$^{-1}$ Hg) as reference material after every 18th measurement and four standards every beginning of a measuring day. The detection limit of the most sensitive cuvette was <0.003 ng Hg. For each sample, we measured THg at least two

times and up to four times if the results showed larger variations.

### 3.4 Diatom analysis

We analyzed diatoms in a total of 54 samples from sediment core EN18218 and 20 samples from surface core 16-KP-04-L19B, taken from 0.5 cm slices. For light-microscopy based species identification we prepared diatom

slides following the procedure described in Battarbee et al. (2001). We treated 0.1 g of freeze-dried sample material with hydrogen peroxide (30 %) for up to 5 hours, added hydrochloric acid (10%) to stop the reaction and washed



the sample with purified water before adding between 5-8x10⁶ microspheres, according to the density of valves on the test slides, to estimate concentration of diatom valves (DVC). Homogenized sediment suspension was transferred to cover slips placed in Battarbee cups to avoid species fractionation, and mounted to slides using

Naphrax. To identify diatoms to the lowest possible taxonomic level we used a Zeiss Axioscope 5 Light microscope with an Axiocam 208 color camera attached, equipped with a Plan-Apochromat 100x/1.4 Oil Ph3 objective at 1000x magnification. We counted more than 300 diatom valves in each sample (Wolfe, 1997) in both sediment cores (mean 351 valves in EN18218; mean 375 valves in 16-KP-04-L19B). Diatom species identification was based on various literature including Hofmann et al. (2011) and Krammer and Lange-Bertalot (1986-1991) as

well as online databases (i.e. http://www.algaebase.org). Correct identification of species was supported by images from a scanning electron microscope and for some species with input from the diatom community online platform DIATOM-L (Bahls, 2015). During diatom analysis, valves were distinguished in pristine and non-pristine valves, and chrysophyte cysts were counted but not further identified.

**3.5 Data processing and statistics**

For statistical analysis of downcore proxy data we used the R environment (R Core Team, 2016). Both cores, EN18218 and 16-KP-04-L19B were statistically analyzed following the same procedure.

To create diatom zones along the cores we used the package 'rioja' for constrained incremental sums-of-squares clustering (CONISS) based on Bray-Curtis dissimilarity after log transformation of all count data to downweigh abundant species (Grimm, 1987).

We used the 'decorana' function in the package 'vegan' (Oksanen et al., 2020) to apply a detrended correspondence analysis (DCA) on percentage data and calculated gradient length in standard deviation units (SD EN18218 = 2.36; SD 16-KP-04-L19B = 1.25). According to the threshold suggested by Birks (2010) we chose principal component analysis (PCA) to reveal major trends in square-root transformed data based on Euclidean distance. Before PCA, we filtered the species data to exclude rare species, i.e. only species present with ≥ 3% in ≥ 2 samples

were included in PCA. The bottom sample at 540 cm was too different from the rest of the more established diatom assemblage and was therefore excluded from the analysis. To downweigh abundant species, filtered data were square-root transformed.

We estimated diatom species richness (alpha diversity) based on Hill's N0, and N2 diversity (Hill, 1973) and performed rarefaction to correct richness estimates for differences of valve counts (Birks et al., 2016) using the

'vegan' R package (Oksanen et al., 2020). The minimum base sum of all samples was n=304 for EN18218 and n=333 for 16-KP-04-L19B.

To estimate diatom valve dissolution, we calculated the *F* index following Ryves et al. (2001), providing a range between 0 and 1 in which 0 is poor and 1 is perfect preservation.

Pearson correlation matrices were generated using the 'cor' function in the R package 'stats'. To highlight

significant correlations by p value criterion, a significance test based on the upper tail probability from the pearson correlation coefficients was performed using the function 'cor.mtest' in the R package 'corrplot'. Only correlations that yielded p values <0.05 were considered significant.

The TOC/$N_{atomic}$ ratios were calculated following Meyers and Teranes (2002) using weight ratios of TOC and N multiplied with 1.167. TOC from EN18218 was used from Vyse et al. (2021) to estimate TOC/N ratios for the

long core.



To calculate individual accumulation rates, we first multiplied dry bulk densities (DBD in g cm$^{-3}$) with sedimentation rates (SR in cm a$^{-1}$) to generate dry mass accumulation rate values (MAR in g cm$^{-2}$ a$^{-1}$). Since the DBD measurement of the 16-KP-04-L19B surface sample was missing, we extrapolated the value from samples below by constructing a piecewise polynomial in the Bernstein basis using the Python package "scipy" (Virtanen

et al., 2020). To determine uncertainty ranges for the individual accumulation rates we propagated the 2σ uncertainty range of SR into the MAR calculations.

Organic carbon accumulation rates (OCARs) were estimated by dividing TOC (wt-%) by a factor of 100, then multiplying with MAR and converting to g m$^{-2}$ a$^{-1}$ units by multiplying it with a factor 10000. We multiplied Hg values with MAR to estimate mercury accumulation rates (HgAR), but then converted HgAR to μg m$^{-2}$ a$^{-1}$ units

by multiplying the HgAR values with a factor 10. Diatom accumulation rates (DAR in valves m$^{-2}$ a$^{-1}$) were estimated following Birks (2010) by using MAR multiplied with the diatom valve concentration (valves g$^{-1}$) and concerting it to 10$^9$ valves m$^{-2}$ a$^{-1}$ units.

Mean summer insolation was calculated for Rauchuagytgyn at (90° from vernal point, 67.8° N, 1365 solar constant) using QAnalySeries 1.5.1 and Earth's orbital parameters from Laskar et al. (2004). Pollen-reconstructed

July temperatures (TJuly$_{pollen}$), and annual precipitation (APP$_{pollen}$) from Andreev et al. (2021) were resampled onto the core depths based on their published ages and our age-model output from EN18218. Mean July temperatures from the closest weather station OSTROVNOE (first observation 1936 CE, 68.12°, 164.17°, ID: RSM00025138, 98 m.a.s.l., 195 km W of Lake Rauchuagytgyn) were estimated from daily values retrieved from NOAA (www.noaa.gov) and resampled onto to sample depths of 16-KP-04-L19B based on the plum age model.

**4 Results**

**4.1 Chronology**

The LANDO age-depth model based on radiocarbon dates for EN18218 (Fig. 3) shows an age range of 28,190 to 29,907 cal yr BP (weighted mean age: 28,950 cal yr BP) at 651.75 cm, which agrees with the age-depth model developed by Vyse et al. (2021) of about 29,000 cal yr BP at core base. The weighted mean sedimentation rates

of all LANDO models ranges over the entire core from 0.01 cm yr$^{-1}$ to 0.1 cm yr$^{-1}$, which means that the maximum value is higher in some regions than previously reported (Vyse et al., 2021) with a maximum of 0.054 cm yr$^{-1}$. However, both models agree on decreases in mean sedimentation rates below 0.02 cm yr$^{-1}$ around 558-560 cm and 346-358 cm, and increases around 510-519 and 371-374 cm, where LANDO models suggest mean values above 0.065 cm yr$^{-1}$. Additional decline in mean sedimentation rate is found below 0.02 cm yr$^{-1}$ approximately between

224-243 cm. Taking into account the 2σ confidence intervals of all models, the uncertainty for the sediment core ranges from minimum values of 0.002 cm yr$^{-1}$ at 569 cm to maximum values of 0.575 cm yr$^{-1}$ between 52-54 cm. The Plum age-depth model based on lead and cesium dates for short core 16-KP-04-L19B (Fig. 4) reaches a maximum mean age of 1864 CE (uncertainty range: 1813 – 1905 CE) at 11 cm. Total $^{210}$Pb activity reached equilibrium with the supporting $^{226}$Ra at a depth of around 7 cm (Fig. 2), which explains the larger uncertainty

between 7 and 11 cm (Fig. 4). Unsupported $^{210}$Pb concentrations vary irregularly with depth with significant non-monotonic features between 1-2.5 cm and 3-4.5 cm. Mean sedimentation rates range between 0.03 cm yr$^{-1}$ (2-3 cm) and 0.139 cm yr$^{-1}$ (0-1 cm), where the higher values can be explained by the lack of $^{210}$Pb data for the first



centimeter. The uncertainty range (2σ confidence interval) for the sedimentation rate over the entire short core lies between 0.025 cm yr$^{-1}$ and 0.192 cm yr$^{-1}$.

## 4.2 Diatom species assemblages


Diatoms occurred upward of 541cm (21.8 cal ka BP) in long core EN18218 (Fig. 5) and were found in all samples between 0 and 10.5 cm counted from short core 16-KP-04-L19B (Fig. 6). In total 204 different species were identified. The dominant taxa in the observed samples are represented by planktonic cyclotelloid species, *Aulacoseira* and small achnanthoid species. Chrysophyte cysts only occurred with few counts in two samples (251

and 311 cm in EN18218) and were therefore neglected. The valve dissolution *F* index in both long (min. 0.78, max. 0.95, mean 0.90) and short core (min. 0.86, max. 0.98, mean 0.94) was generally high referring to an overall good valve preservation.

Mean diatom valve concentrations (DVC) in EN18218 were 46.4 (2.3-134.7) 10$^7$ valves g$^1$, corresponding to diatom valve accumulation rates (DAR) of 176.1 (2.0-651.6) 10$^9$ valves m$^2$ a$^1$. Modern DVC in the short core had

a relatively higher mean of 82.0 (43.8-200.4) 10$^7$ valves g$^1$, corresponding to DAR of 199.2 (42.2-514.9) 10$^9$ valves m$^2$ a$^1$.

Rarefied species richness Hill's N0 varied between 11.8 and 42.2 (mean 29.4) in the long core and was slightly higher between 31.1 and 47.4 (mean 38.8) in the short core. The effective richness Hill's N2 ranged between 1.5 and 8.4 (mean 3.1) in the long and between 2.9 and 6.1 (mean 4.3) in the short core. A remarkable shift toward

high effective richness is found in the long core between ca. 241 and 346 cm.

The first three directions in the principal component analysis explain ca. half of the data variance in EN18218, i.e. PC1, PC2, and PC3 explained 23.2, 17.8, and 11.8 %, respectively. The first three PCA axes from the short core assemblage data explained ca. two third of the data variance, i.e. 28.9, 22.8, and 14.4 % for PC1, PC2, and PC3, respectively. PC3, however, was not included in the biplots shown in Fig. 7.


According to our cluster analysis we divided the cores into 7 diatom zones A-G in core EN18218 and 3 zones H-J in the surface core 16-KP-04-L19B and described the species in chronological order.

*Diatom Zone A 541-505 cm (EN18218, 21.8-20.2 cal ka BP)*

The oldest diatoms found in the core were dominated by planktonic *Lindavia ocellata* (~57.7%), *Lindavia cyclopuncta* (~26.2%) and *Lindavia bodanica* (~3.1%), accompanied by benthic *Achnanthidium minutissimum*, *Encyonema minutum* and *Nitzschia palea* with lesser abundance.

*Diatom Zone B 505-426 cm (EN18218, 20.2-15.3 cal ka BP)*

Zone B was dominated by *L. cyclopuncta* (~49.7%), *L. ocellata* (~33.6%), and, appearing in two samples only, *Aulacoseira valida* (~2.5%). *Pliocaenicus costatus* occurred frequent at 441 cm (7.9 %). Benthic species *Staurosira construens*, *Encyonopsis descriptiformis*, *Psammothidium chlidanos* and *Hannaea arcus* appeared with low abundance in addition to the benthic species found in zone A.

*Diatom Zone C 426-366 cm (EN18218, 15.3-12.8 cal ka BP)*



Dominant planktonic species were represented by *L. ocellata* (~44.1%), *L. cyclopuncta* (~32.4%) and *P. costatus* (~4.9%). *Staurosira pinnata* and *Staurosira brevistriata* occurred in addition to the benthic species found in zone B.

*Diatom Zone D 366-346 cm (EN18218, 12.8-11.4 cal ka BP)*

Zone D appeared as a small section (two samples) in which only L. ocellata remained frequent (~61.6%), while other planktonic forms were restricted to first-time occurrence of *Aulacoseira subarctica* (~4.5%) and less frequent *P. costatus*. *Brachysira neoexilis* (~4.1%) started to occur while *S. pinnata* started to became more frequent (~3.0%).


*Diatom Zone E 346-241 cm (EN18218, 11.4-8.0 cal ka BP)*

In the Early Holocene part of the core dominant planktonic species were *L. cyclopuncta* (~37.7%) and *L. ocellata* (~14.8%), accompanied by generally more frequent benthic forms represented by *S. pinnata* (~7.9%), *B. neoexilis* (~6.8%), and *A. minutissimum* (~3.8%).


*Diatom Zone F 241-166 cm (EN18218, 8.0-5.3 cal ka BP)*

Zone F started with the dominance of *L. cyclopuncta* (~57.7%), while other *Lindavia* species disappeared. Instead, *A. subarctica* (~4.8%), *P. costatus* (~3.2%) and *A. valida* (~2.9%) occurred in higher frequencies. *S. pinnata* reached highest values (~8.4%) while *P. chlidanos* and *H. arcus* also showed peaking values.


*Diatom Zone G 166-11 cm (EN18218, 5.3-1.1 cal ka BP)*

The upper part of long core EN18218 was characterized by the re-occurrence of *L. ocellata* (~15.0%) and strong representation of *A. subarctica* (~9.2%), while *L. cyclopuncta* (~53.8%) remained the dominant species. Benthic *Staurosira* forms and *A. minutissimum* decreased.


*Diatom Zones in the short core H-J 10.5-7.0-1.5-0.5 cm (16-KP-04-L19B, 1870-1907-2003-2012 CE)*

The strongest shift within the species assemblage of the short core is found at 7 cm. Overall, *L. cyclopuncta* (~37.3%) and *A. subarctica* (~24.4%) remain the most dominant species. *L. ocellata* appears with highest abundance in zone I, i.e. between 7.5 and 4.5 cm (~17.5%), while the same zone is characterized by high values

of benthic *Pinnularia nodosa*.

**4.3 Biogeochemical variables**

We complemented geochemical data from core EN18218 (Fig. 8) provided by Vyse et al. (2021) for TN and THg measurements. TN varied from >0.1 to 0.25 wt%, with highest values in the upper 100 cm of the core. Resulting TOC/TN$_{atomic}$ ratios ranged between 6.0 and 19.2, with strong increase at 341 cm. THg in the same core ranged

between 93.2 and 362.8 µg kg$^{-1}$, with highest value in the sample at 600.25 cm and overall higher mean values above 321 cm (mean 198.6 compared to 141.8 µg kg$^{-1}$ below). Mean Hg accumulation rates (HgAR) estimated from these concentrations were 63.4 (11.8-138.6) µg m$^{-2}$ a$^{-1}$. Mean organic carbon accumulation rates (OCAR) estimated from TOC values published by Vyse et al. (2021) were 4.6 (0.8-12.7) g m$^{-2}$ a$^{-1}$. Low peaks in OCAR,





DAR, and HgAR in both cores correspond to low sedimentation rates at 230, 350, and 550 cm in EN18218, and between 2 and 3 cm in 16-KP-04-L19B (Fig. 3 and 4).

Data from short core 16-KP-04-L19B are presented for the first time (Fig. 6 and 9). TOC ranged between 2.6 and 3.5 wt%, with highest values in the upper 3 cm. N varied between 0.28 and 0.41 wt%, resulting in TOC/TN$_{atomic}$ ratios with only little fluctuation around mean 10.7, which is fitting to the upper part of EN18218. THg in the short core varied between 162.4 and 244.7 µg kg$^{-1}$, with highest values in between 4.5 and 3 cm. Mean OCAR and

HgAR estimated from TOC and THg concentrations were 6.7 (2.7-11.5) g m$^{-2}$ a$^{-1}$ and 46.0 (14.3-69.8) µg m$^{-2}$ a$^{-1}$, respectively.

## 5 Discussion

### 5.1 Ecological responses of diatom species to Late Quaternary environmental changes

Diatoms in Lake Rauchuagytgyn started to appear at 21.8 cal ka BP (Fig. 5) with strong dominance of *L. ocellata*

(Pestryakova et al., 2018), a planktonic and ultraoligotrophic to mesotrophic taxon, common in cold lakes (Wunsam et al., 1995). The first occurrence of diatoms was accompanied with the first increase of organic carbon accumulation (Fig. 8). According to previous sedimentological work on the sediment core (Vyse et al., 2021), at this time glaciers retreated from the catchment and unfrozen episodes became more frequent leading to paraglacial deposition progressing in the lake basin. In the course of continued deglaciation since ca. 20 cal ka BP in Chukotka

(Vyse et al., 2020) and Alaska (Elias and Brigham-Grette, 2013) the diatom assemblage developed progressively in diatom zones A and B, enabling oligotrophic *L. cyclopuncta* (Scussolini et al., 2011) and few benthic species to occupy ecological niches in the young and still cold lake ecosystem. Strong fluctuations, e.g. of tychoplanktonic *A. valida*, still indicates unstable habitat conditions. *Pliocaenicus costatus* is known in larger quantities from cold and strongly oligotrophic mountain lakes restricted to East Siberia (Cremer and Van De Vijver, 2006). Limitation

of benthic diatoms may result from in-wash of clay during deglaciation (Vyse et al., 2021) leading to low-transparent and narrow littoral zones in an overall deep basin. In diatom zone C the species richness increased strongly and benthic diatoms became abundant (Hill's N0, planktonic/benthic ratio in Fig. 8) supporting a gradual climate amelioration equivalent to the Bølling-Allerød interstadial, which started ca. 15.5-15.0 cal ka BP (Wohlfarth et al., 2007; Obase and Abe-Ouchi, 2019; Andreev et al., 2021) facilitating shallow water habitats and

thus more complex diatom communities due to longer growing seasons (Cherapanova et al., 2007).

The short but remarkable diatom zone D is characterized by the same cold-adapted planktonic and parts of benthic species from the early deglacial period in diatom zone B. Thus, in accord to other findings from Chukotka (Anderson and Lozhkin, 2015) and i.e. Lake El'gygytgyn (Andreev et al., 2012), the Rauchuagytgyn diatom assemblage provides evidence of an aquatic ecosystem response to climate cooling and drying between ca. 12.8-

11.4 cal ka BP corresponding to the Younger Dryas.

The Pleistocene/Holocene (P/H) boundary is detected from the diatom assemblage change at ca. 346 cm in core EN18218, fitting well to the uncertainty range of 10.8-12.2 cal ka BP (Fig. 5 and Fig. 3). At the glacial-interglacial transition, the diatom community responded with a strong decrease of planktonic and light *Lindavia* species (Biskaborn et al., 2021b) that was accompanied with a decrease in both diatom and carbon accumulation rates

(Fig. 8). Mountain ice-sheets that persisted in the catchment over the deglacial period vanished at the P/H boundary leading to decreased water supply and lower lake levels over the Early Holocene. At that time, the effective species



richness (Hill's N2) increased because relatively more benthic species reached higher percentages, while the pure richness (N0) slightly decreased. The P/H is also characterized by a clear increase of the first axis sample scores of the PCA (Fig. 8). The PCA biplot depicts grouping of planktonic *Lindavia* versus benthic *Staurosira* and

*Psammothidium* species along the primary axis while *Aulacoseira* species are oriented along the secondary axis (Fig. 7). Fragilarioid forms such as *S. pinnata*, *S. construens*, and *S. brevistriata* are known as typical small benthic pioneering forms in boreal shallow lakes (Valiranta et al., 2011; Biskaborn et al., 2012) that are often alkalophilous (Paull et al., 2017). Together with the increase of achnanthoid taxa, this assemblage indicates increased availability of littoral habitats with nearby woody vegetation (Andreev et al., 2021) on fresh soils leading to enhanced ion

supply and eventually increased alkalinity to the lake (Herzschuh et al., 2013).

At the P/H transition, abrupt high TOC/TN$_{atomic}$ values of around 15 (Fig. 8) point to a higher contribution of less-degraded organic carbon, indicating an increase of benthic water plants and terrestrial plant material (Meyers and Teranes, 2002; Baird and Middleton, 2004) and thus provides evidence for shallower shores and development of catchment vegetation due to maximal summer insolation and warm interglacial conditions (Fig. 8). The increased

role of water plants is supported by epiphytic *B. neoexilis*, *E. minutum*, *A. minutissimum* and *E. descriptiformis* (Barinova et al., 2011; Hofmann et al., 2011).

Swann et al. (2010) reconstructed most favorable climate conditions known as the Holocene Thermal Maximum (HTM) at Lake El'gygytgyn (140 km to E) between 11.4 and 7.6 cal ka BP. However, based on pollen data, Andreev et al. (2021) reconstructed the start of warmest conditions at ca. 8.0 cal ka BP for the Rauchuagytgyn

region. At 8.0 cal ka BP in diatom zone F, *L. ocellata* disappeared together with low levels of *L. bodanica,* while *Aulacoseira* species started to establish. *Aulacoseira* builds heavy and rapidly sinking frustules commonly found in deep boreal lakes (Laing and Smol, 2003). Euplanktonic *A. subarctica* is a pelagic species that requires turbulence to remain in the photic zone (Rühland et al., 2008; Gibson et al., 2003), while light cyclotelloid taxa prefer stratified water conditions (Rühland et al., 2015). We assume that high July temperatures continued but in

addition the open water seasons prolonged around 8-7 cal ka BP, as winters in Siberia gradually became warmer over the mid- and Late Holocene (Meyer et al., 2015). Thus, early ice-out and longer spring circulation supported *Aulacoseira* species (Horn et al., 2011) and led to a drastic change in the Rauchuagytgyn species assemblage. For comparison, in recent times, a surface ice-layer in Chukotka lakes builds up in October reaching up to 1.8 m over the winter, and breaks up in early July after snow melt started in mid-May (Nolan et al., 2002).

At ca. 6.4 cal ka BP the DAR, OCAR, TOC/TN$_{atomic}$ ratios increased (Fig. 8), while *L. bodanica* reappeared, *A. valida* increased strongly but small benthic *Staurosira* species retreated (Fig. 5). We interpret this change with maturation of soils (Biskaborn et al., 2012), retreating woody vegetation (Andreev et al., 2021), and a shift in bioproduction driven by increased supply of nutrients through increased river activity and increasing water levels (Buczkó et al., 2013). During the cooling of the Late Holocene the diatom zone G assemblage continuous as a

semi-pelagic cold water community at intermediate to high water levels, high DAR but slightly decreased richness and also decreasing terrestrial influence indicated by lowering TOC/TN$_{atomic}$ values.

The last decades are represented in the short core 16-KP-04-L19B about 200 m E of the long core position. These surface sediments were also dominated by the same *Lindavia* and *Aulacoseira* species as compared to diatom zone G (Fig. 9). The main assemblage change occurred at ca. 1907 CE characterized by increases in the benthic taxa

*Psammothidium chlidanos* and *Pinnularia nodosa*, accompanied by a slight shift from *Aulacoseira* to *Lindavia* species and decreasing OCAR, DAR, and HgAR, but slightly increased TOC/TN$_{atomic}$ values (Fig. 9). The *Aulacoseira-Lindavia* shift is tentatively supported in the PCA biplot in PC1 (Fig. 7). Even though the overall



response to recent environmental changes in Rauchuagytgyn seems of minor extent, the timing and response corresponds to warming at high latitudes observed during industrialization (Biskaborn et al., 2021a). Abrupt shifts

in lake ecosystems were most frequently observed around 1950 CE (Huang et al., 2022) when the beginnings of human energy consumption and geochemical impacts initiated the (proposed) Anthropocene epoch (Syvitski et al., 2020). A strong warming at 1950 CE was also documented in air temperature observations in the weather station 195 km W of the study area. In Rauchuagytgyn, DAR decreased strongly at that time, while HgAR and TOC/TN$_{atomic}$ ratios fluctuated having a negative influence on species richness N0 and N2 (Fig. 9). In between ca.

1960 and 1985 CE there is a peak in *Tabellaria flocculosa*, a species that can occur with both planktonic and benthic life styles (Heudre et al., 2021) indicating slightly acidic environmental conditions, and may be responding to unstable habitat conditions (Palagushkina et al., 2012). Since 1990 CE DAR, OCAR and Hg increased again while *A. subarctica* and *P. nodosa* decreased, pointing to either minor atmospheric influences on the lake hydrochemistry, or natural short-term variation (Gibson et al., 2003).


**5.2 Correlation between carbon, diatoms and mercury accumulation**

Accumulation rates (AR) are generally rather uncertain, due to limitations in precise estimation of sedimentation rates from age-model interpolations (Sadler, 1981). In our study this effect is relevant due to the conversion from weight concentrations of organic carbon and mercury to AR through multiplication with sedimentation rates from

age modelling and bulk densities related to water contents (Vyse et al., 2021). Diatom concentrations, however, are expressed as numbers of frustules (Battarbee et al., 2001). Accordingly, one cannot directly infer diatom biomass from count-based DAR as taxa vary considerably in size among and within species (Birks, 2010). As discussed above, the diatom succession in the Rauchuagytgyn sedimentary record shows a tendency toward lake ontogeny (Brenner and Escobar, 2009). We thus expect (unknown) changes in the relation between biomass and

number of valves that cannot be ruled out. Acknowledging the overall strong bias in AR we still attempt to quantify carbon, diatom, and Hg trajectories to help assessing northern lake systems as freshwater carbon sinks.

At millennial time scale in the long core OCAR is strongly correlated with HgAR (r 0.94, p<0.05) and significantly with DAR (r 0.64, p<0.005), while there is no significant correlation between diversity indices to HgAR (Fig. 10). The mean values at around 11.2 (and 12.9 in the Holocene) of TOC/TN$_{atomic}$ measured in the long core represent

a mixture of (in simple words) high-N planktonic algae, medium-N benthic water plants and low-N terrestrial vegetation input (Baird and Middleton, 2004). Phytoplankton produces TOC/TN$_{atomic}$ ratios of 4-10, whereas vascular land plants produce >20 (Meyers and Teranes, 2002). Given the sparse vegetation cover around the lake (Huang et al., 2020) and the overall low TOC/TN$_{atomic}$ ratios terrestrial input may play a minor role. We therefore assume that there is a strong contribution of algae to the bulk organic matter accumulated in the lake which tends

to be somewhat proportionate to the number of diatom valves. Accordingly, the TOC/TN$_{atomic}$ ratios correlate negatively with planktonic/benthic ratios in the long core (r -0.57, p<0.05), but only slightly (insignificantly) in the short core (r 0.21, p>0.05). This mismatch could indicate that there is anthropogenic nitrogen contribution from the atmosphere (Biskaborn et al., 2021a) that is in addition masked by short-term fluctuations and constraints of measurement precision in high-resolution samples from lakes under extreme environmental conditions. A

tentative relationship between DAR and planktonic species could be detected in the short core (r 0.41, p<0.05)





which could indicated, that the wide spread increase of planktonic species in high-latitude lakes as a response to global warming (Smol et al., 2005) contributes to increased diatom primary productivity.

Over the last centuries visible in the short core there is also a significant correlation between OCAR and HgAR. Atmospheric mercury, however, is not simply deposited in Arctic lakes, but instead there is a strong influence of
limnological processes such as primary production and ice cover dynamics on mercury biogeochemical cycling (Korosi et al., 2018). The correlation between OCAR and DAR (r 0.65, p<0.05) apparently shows that diatoms play a role in these processes. In contrast to long time scales there is a significant negative correlation in the short core between HgAR and diversity estimates Hill's N0 (r -0.51, p<0.05) and N2 (r -0.39, p<0.05). Thus, contaminants during the industrial period could be assumed to have a stronger effect on the lake ecosystem than
natural Hg supply before increased anthropogenic activity (Huang et al., 2022). Studies on deep permafrost soils in Siberia showed that average Hg concentrations of 9.7 µg kg$^{-1}$ could be used as a baseline for natural Hg concentrations (Rutkowski et al., 2021). However, as Hg binds to lake organic carbon (Braaten et al., 2018), lake bioproductivity is likely increasing the mercury load within sediments, explaining the overall high concentrations in older sections. Furthermore, we found a very good correlation between HgAR and OCAR during the cold glacial
period (r 0.98 p<0.05) but obvious decoupling from diatoms as shown by missing correlation between HgAR and DAR (Fig. 10). It can be assumed that other C sources than within-lake bioproduction, such as external supply of organic matter represented the main source of the cold climate carbon deposition.

**5.3 Long-term ecosystem feedbacks to climate changes**

Well-preserved and old diatom records in Chukotka provide the opportunity to study direct responses of natural lake ecosystems to regional climate changes (Cherapanova et al., 2007; Swann et al., 2010). The Lake Rauchuagytgyn sediment record provides insight to compositional changes of diatom assemblages in response to lake and catchment changes. The main changes observed are best represented by shifts within planktonic species and their proportions relative to benthic forms populating emerging habitats. Significant negative correlations were
found at millennial time scales between planktonic-benthic ratios and diversity estimates Hill's N0 (r -0.7, p<0.05) and N2 (r -0.53, p<0.05), indicating that long-term diatom diversity in Lake Rauchuagytgyn was closely related to lake ontogeny. The general catchment maturation accompanied by decreases of glacial ice-sheet influence led to initiation of new ecological niches and thus diversification of e.g. epiphytic species (Wilson et al., 2012; Rouillard et al., 2012).

Diatom accumulation rates on the other hand, recorded independently from lifestyles, show a clear relationship to lake bioproduction. Relationships between diatoms and organic carbon in lake sediments were also related to species alpha diversity in lake Bolshe Toko (Biskaborn et al., 2021b) and explained as stabilizing effects of well-developed species richness supporting primary biomass production (Marzetz et al., 2017). The correlation found between diatom valve and organic carbon accumulation rates during the interglacial in the Rauchuagytgyn
sediment record may support that, during warm episodes, diatoms in high-latitude lakes with relatively small catchments are coupled with the bulk production of biomass (Figure 11). Whether or not diatoms might also be synchronized with catchment production remains unclear but not unlikely. Nevertheless, in Lake Rauchuagytgyn we observed a positive feedback mechanism of the freshwater diatom biomass to long-term climate amelioration and thus an increasing role of Arctic lakes as sink for organic matter (Hughes-Allen et al., 2021; Vyse et al., 2021).



Compared to ocean systems, where fertilization projects attempted to force carbon burial by artificial diatom blooms (Yoon et al., 2018), lakes may possess a higher potential to withdraw carbon from the atmosphere because of lower carbon remineralization rates (Mendonça et al., 2017; Sobek et al., 2009). However, this may nowadays be questioned because whole-lake experiments and models suggested possible lagged response of lakes natural resistance to anthropogenic stressors that could cause fast ecosystem switches (Pahl-Wostl, 2003). In turn, these potential alterations could possibly prevent carbon sink feedbacks as observed in the remote and still pristine Rauchuagytgyn system. In this context, the observed positive relationship between sedimentary carbon and mercury suggests potential mitigation feedbacks of contamination stress accompanied by recent climate change. However, impacts of human-driven atmospheric stressors only seem little pronounced in the short core data, which is limiting possibilities to assign natural long-term mechanisms to present day conditions. This is amplified by the fact that northern remote lakes have already passed important ecosystem thresholds and are believed to not represent pristine ecosystems anymore (Smol et al., 2005).

## 6 Conclusions

Radiocarbon and Pb-210 dated sediment cores from Lake Rauchuagytgyn in the Far-East Russian Arctic provide valuable archives of millennial to decadal scale lake ecosystem responses to regional environmental forcing of the last 22000 years before today. Our main findings based on diatom species, organic carbon and nitrogen, and mercury analyses can be highlighted as follows:

- The Pleistocene diatom species assemblage reflects a planktonic community in a deep and cold lake with short growing seasons. The assemblage becomes more complex during a gradual climate amelioration at ca. 15 cal ka BP, similar to Bølling-Allerød, leading to successive development of benthic habitats. Diatom species temporarily returned to glacial conditions between ca. 12.8-11.4 cal ka BP corresponding to the Younger Dryas.
- The Early Holocene diatom community reflects a shallower lake with increased water plants in larger littoral zones and higher alkalinity that we relate to woody vegetation development in the catchment, supported by high carbon to nitrogen ratios. Gradual increasing *Aulacoseira* taxa indicate that winters became warmer over the mid- and Late Holocene leading to earlier ice-out and longer spring circulation.
- During Late Holocene cooling, small benthic *Staurosira* taxa retreated due to soil maturation and increased water levels facilitating higher abundancy of planktonic *Lindavia* and *Aulacoseira* species.
- The last decades represented in a Pb-210 dated short core only show vague evidence of recent change of the diatom community at 1907 CE, indicated by slight increase of light *Lindavia* and decrease of *Aulacoseira* species, accompanied by shifts in the benthic community. Biogeochemical variables and diatom indices fluctuated strongly around 1950 CE.
- Diatom accumulation rates (DAR) and organic carbon accumulation rates (OCAR) do not correlate during the cold episode, but show significant correlation during the warm interglacial when insolation was higher. The Rauchuagytgyn data suggest that during the Holocene (1) deposition of organic carbon was largely driven by within-lake bioproduction, and that (2) diatoms reflect the activity of the gross primary producers of the lake.



- Mercury accumulation rates (HgAR) in the investigated sediments are strongly correlated to OCAR in both cold and warm episodes. As Hg accumulation is bound to organic matter, increased carbon sedimentation during warm climates and suitable biochemical substrate conditions facilitates Hg deposition.
- Pristine boreal lake systems potentially can serve as both $CO_2$ sinks driven by climate-enhanced within-lake primary productivity, and disposal sites for heavy metal contaminants. Consequently, maintaining intact natural lake ecosystems should receive a high priority in future environmental policy.

**Data Availability**

Datasets used in this study will be accessible on PANGAEA close upon publication.

**Acknowledgement**

This project was funded by the European Research Council (ERC) under the European Union's Horizon 2020 Research and Innovation Program (Grant Agreement No. 772852, 'Glacial Legacy'). We thank Justin Lindeman for help in the laboratory related to mercury, carbon, and nitrogen.

**Author contributions**

BKB conceived the study, conducted field work, statistical analyses, and wrote the manuscript. AF performed diatom analysis and counting. GF performed age-depth modelling. LAP, KS-F, and UH coordinated field work and dating of the short core. JS performed mercury analysis. TK performed correlation with p value adjustment. All authors contributed to generating data, writing and review of the manuscript.

**Competing interests**

The authors declare no conflict of interests.

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



**Figures and Tables**

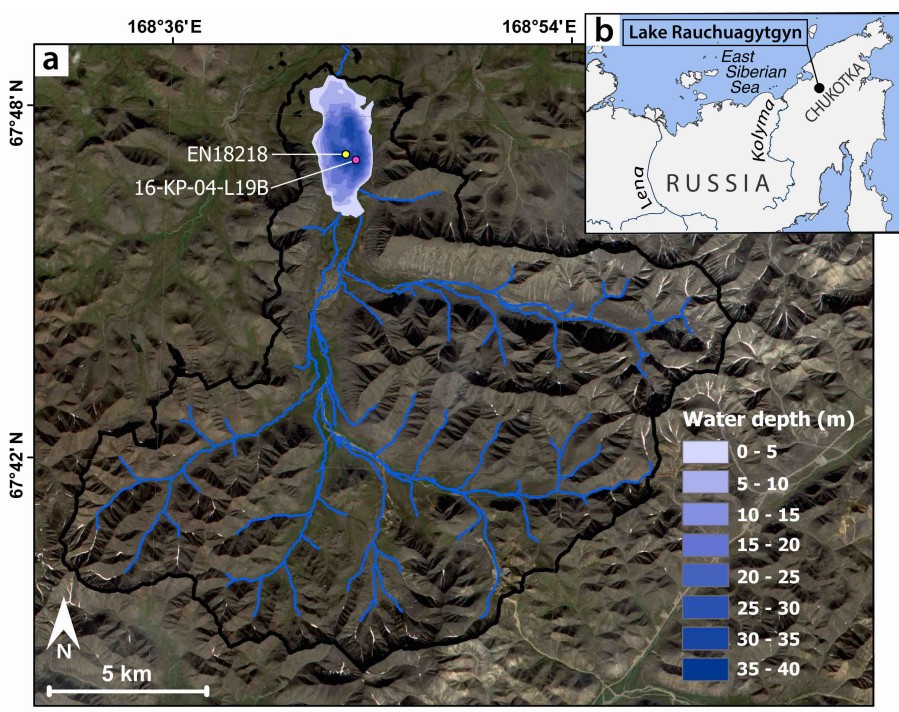

**Figure 1: Study site. a, bathymetrical map of Lake Rauchuagytgyn with catchment area (boundary as**

**black line, inflows as blue lines,) and coring locations (long core EN18218; short core 16-KP-04-L19B). b,**

**geographical overview map. Map based on ESRI (Esri and Geoeye, 2019).**

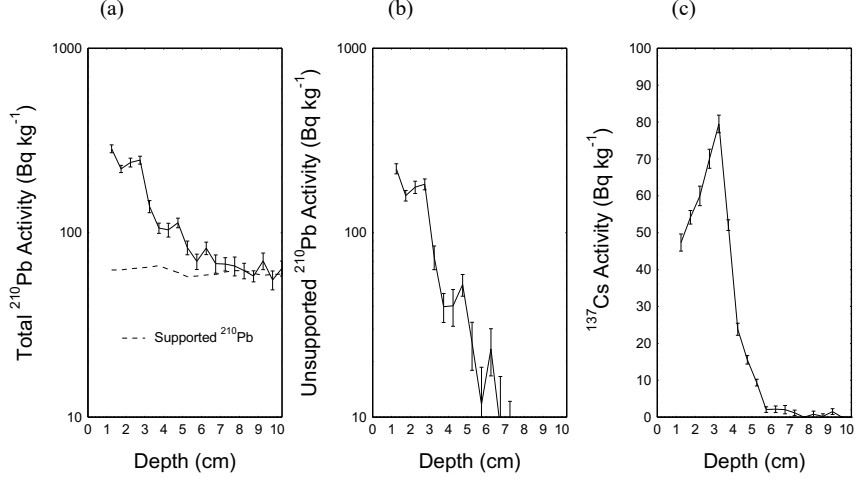

**Figure 2: Fallout radionuclides in short core 16-KP-04-L19B showing (a) total and supported $^{210}$Pb, (b)**

**unsupported $^{210}$Pb, (c) $^{137}$Cs concentrations versus depth.**

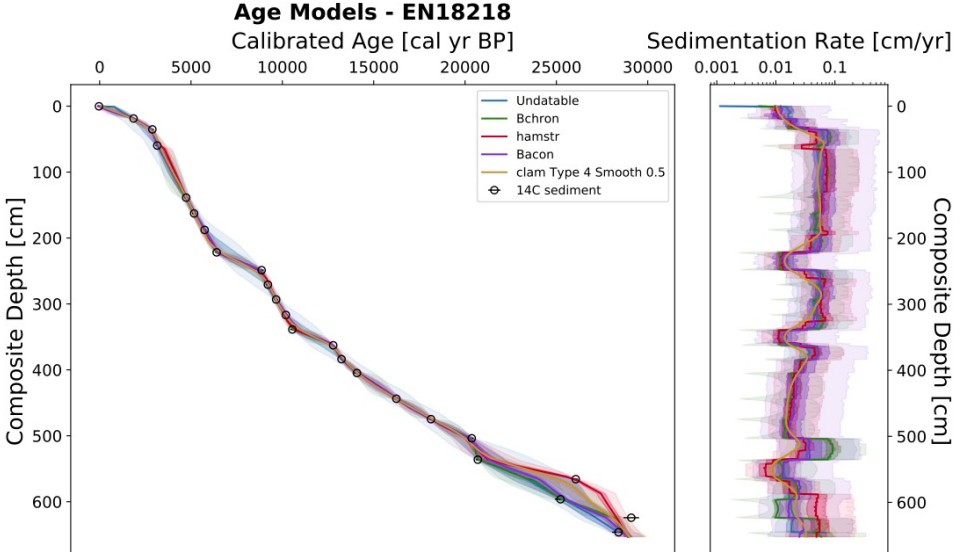

**Figure 3: Generated output from LANDO for sediment core EN18218 based on $^{14}$C data from Vyse et al. (2021). Left plot consists of a comparison between five age-depth models from different modelling codes indicated in the legend. Colored solid lines indicate the median age, while shaded areas represent their respective 1σ and 2σ ranges in the same colors with decreasing opacities. Right plot shows the calculated sedimentation rate with matching colors. Black circles in the left plot indicate the mean calibrated ages of $^{14}$C bulk sediment samples based on IntCal20 calibration curve (Reimer et al., 2020) and their 1σ uncertainty error bars.**



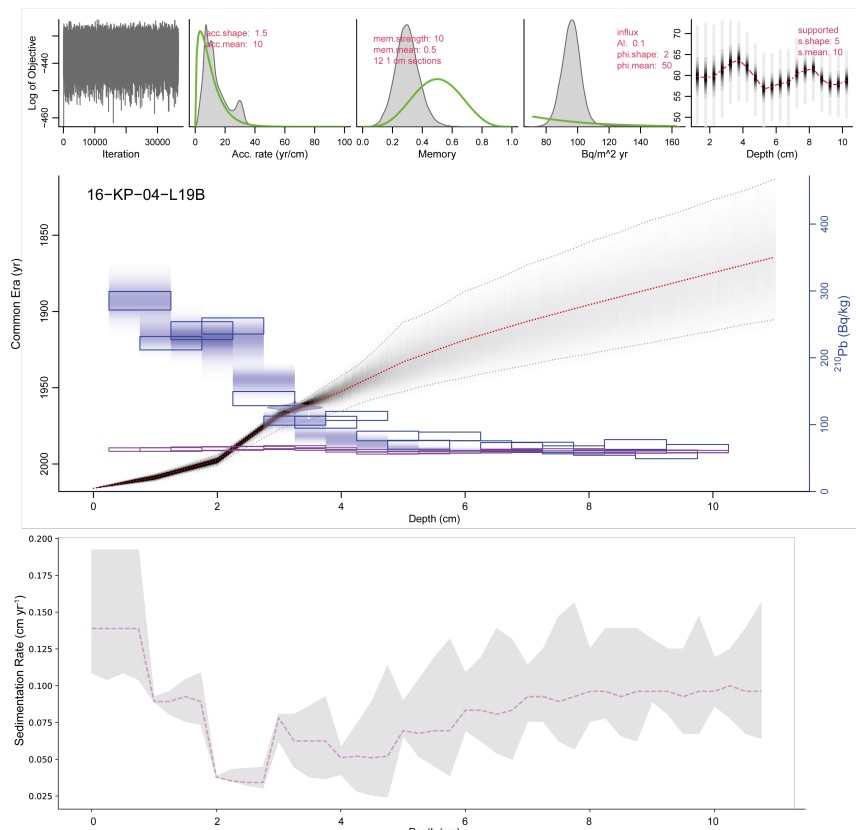

**Figure 4: Plum age-depth model for sediment core 16-KP-04-L19B. The five upper panels show the Bayesian input parameters and their posterior distributions for Plum. The middle panel consists of the age-depth model with its mean age in red and its 2σ confidence interval in grey, the unsupported ²¹⁰Pb concentrations (in Bq/kg) in blue with its 1σ uncertainty, and the supported ²¹⁰Pb concentrations (in Bq/kg) in violet. The lower panel displays the mean sedimentation rate over depth as dashed line and the 2σ confidence interval in grey.**





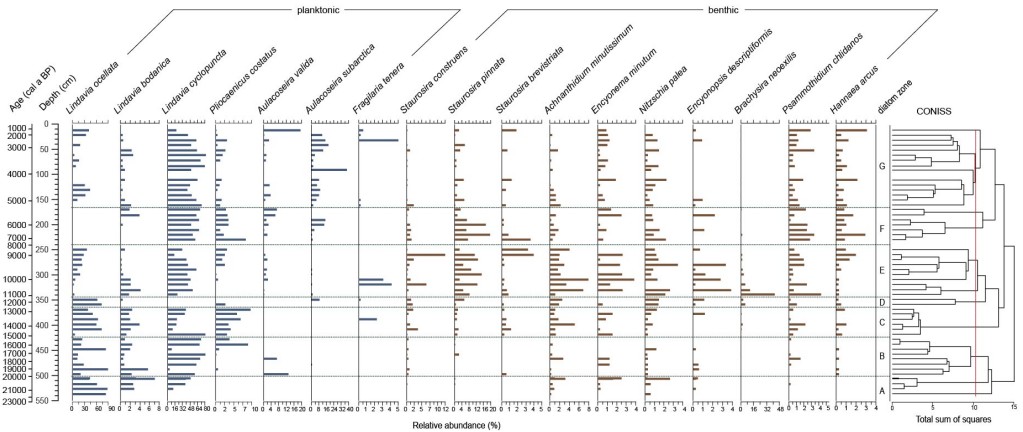

**Figure 5: Relative abundance of diatom species in the long core EN18218. Diatom zones established by CONISS clustering. Species % values are shown next to the mean calibrated ages before present and the core depth below the sediment surface. Taxa present with ≥ 3% in ≥ 2 samples were included in the graph.**


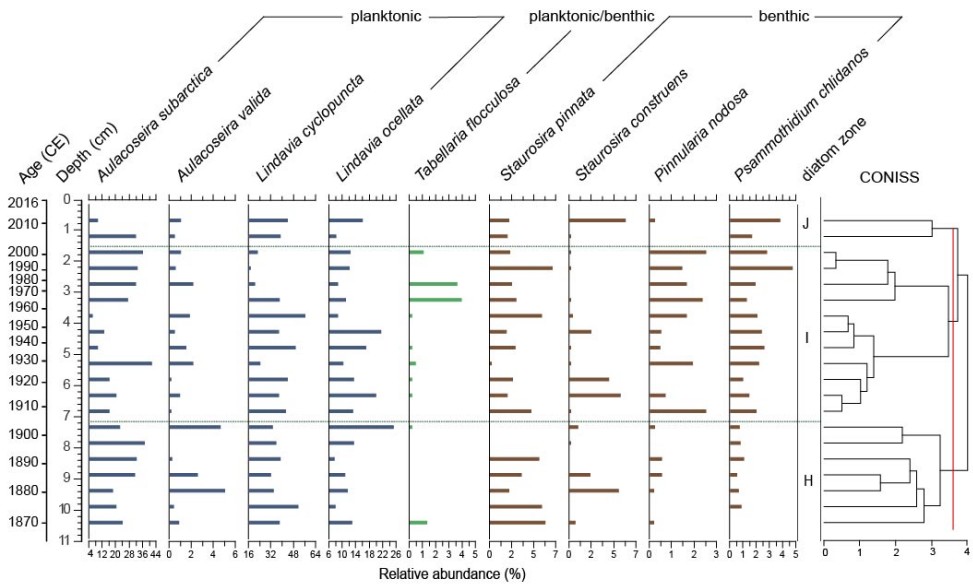

**Figure 6: Relative abundance of diatom species in the short core 16-KP-04-L19B. Diatom zones established by CONISS clustering. Species % values are shown next to calibrated mean ages (common era years, CE) and the core depth below sediment surface. Taxa present with ≥ 3% in ≥ 2 samples were included in the graph.**





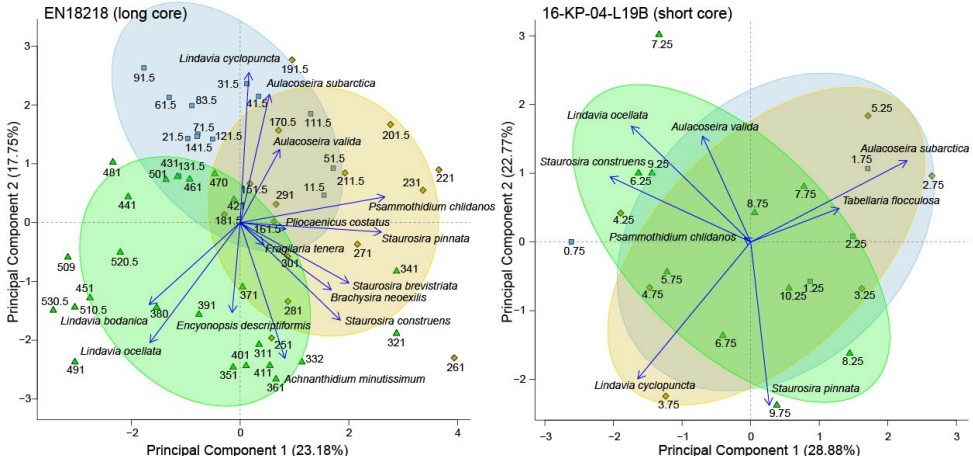

**Figure 7: Biplots of the first two dimensions (PC1, PC2) generated by principal component analysis of**
**diatom species filtered to ≥ 3% in ≥ 2 samples from long core EN18218 and short core 16-KP-04-L19B.**
**Color circles represent eco-taxonomical clusters with comparable environmental preferences. Explained**
**variance of each PC is indicated at the axis label in %.**

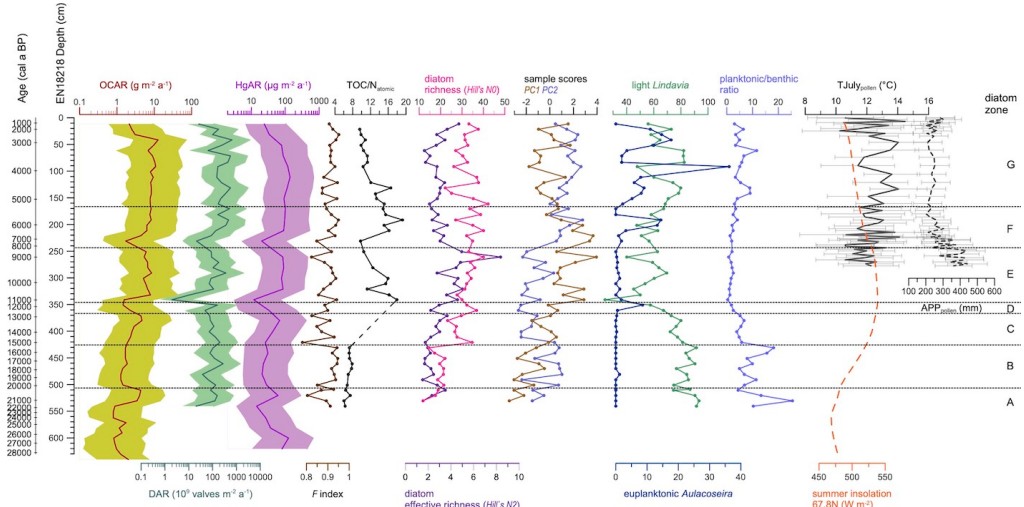


**Figure 8: Biogeochemical variables and statistical diatom indices since the Late Pleistocene from Lake**
**Rauchuagytgyn sediment core EN18218. OCAR, organic carbon accumulation rates; DAR, diatom**
**accumulation rates; HgAR, mercury accumulation rates; *F* index, diatom valve preservation index;**
**TOC/TN_{atomic}, total organic carbon to total nitrogen ratio; diatom species richness based on Hill numbers;**
**PC1-3, main axes sample scores from the principal component analysis; light 'cyclotelloid' *Lindavia* and**
**euplanktonic *Aulacoseira* as sum percentages; planktonic to benthic species ratios; paleoclimate**
**information from published literature at the right. Reconstructed July temperatures and precipitation**



**adopted from Andreev et al. (2021), OCAR and TOC/TN$_{atomic}$ is based on organic carbon concentrations from Vyse et al. (2021), insolation is based on orbital parameters from Laskar et al. (2004).**


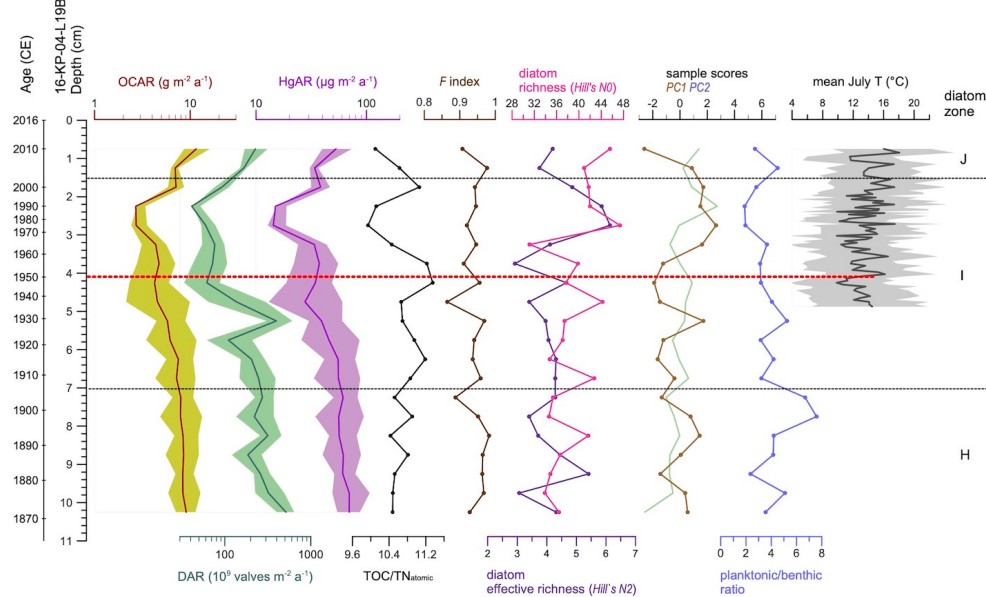

**Figure 9: Biogeochemical variables and statistical diatom indices over the last decades from Lake Rauchuagytgyn surface sediment core 16-KP-04-L19B. OCAR, organic carbon accumulation rates; DAR, diatom accumulation rates; HgAR, mercury accumulation rates; TOC/TN$_{atomic}$, total organic carbon to nitrogen ratio; $F$ index, diatom valve preservation index; diatom species richness based on Hill numbers; PC1-3, main axes sample scores from the principal component analysis; planktonic to benthic species ratios; July temperatures (mean, min, max) calculated from OSTROVNOE weather station (www.noaa.gov). Red dotted line indicates T increase and onset of the Anthropocene**






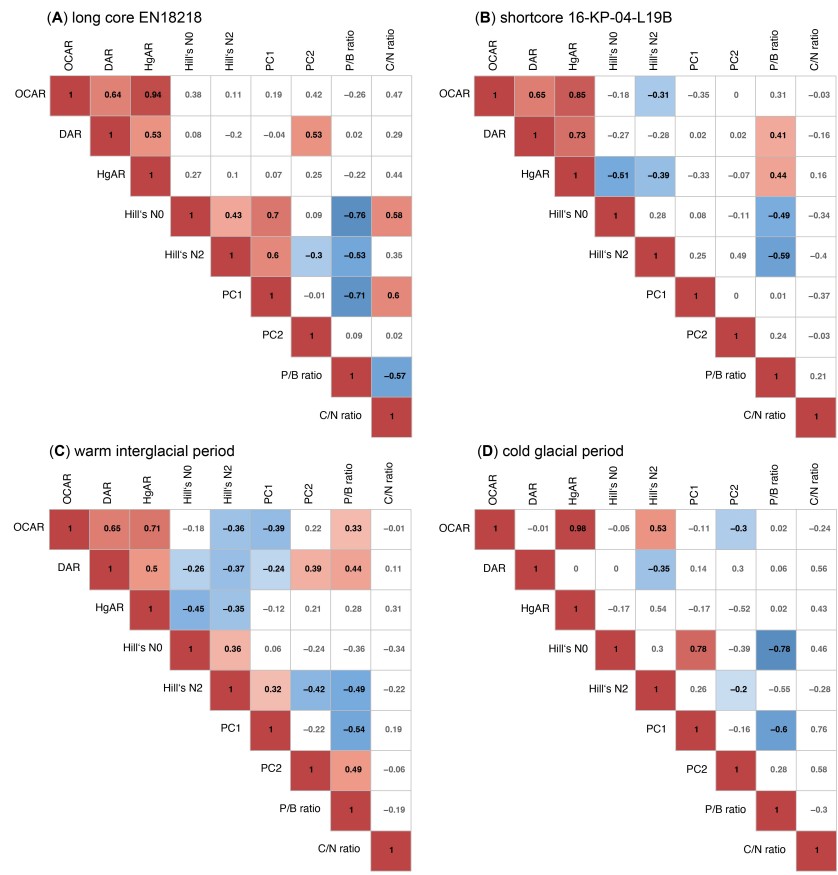


**Figure 10: Pearson correlation matrix between organic matter, diatom indices, and mercury variables. Red, positive correlations; blue, negative correlations. p values correction of <0.05 was applied to assign colors only to significant correlations. Insignificant correlations shown as grey numbers within white cells. Panels A and B represent correlations within the individual core records. Panels B and C represent**

**correlations in the warm period including the short core and the Holocene part of core EN18218 (C), and the cold period restricted to the pre-Holocene part of core EN18218 (D).**



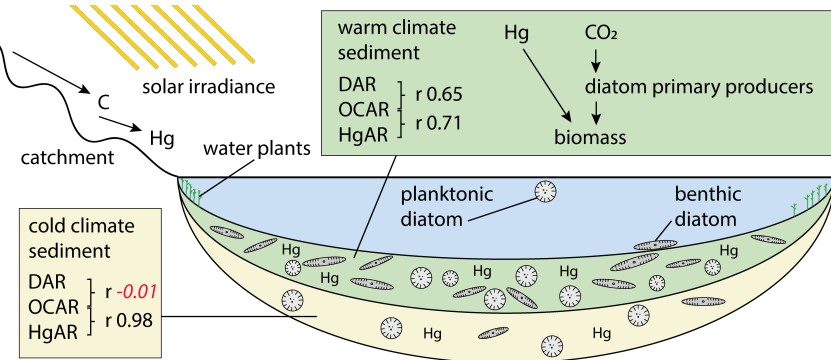

**Figure 11: Schematic drawing of the long-term processes leading to accumulation of diatom valves (DAR), organic carbon (OCAR), and mercury (HgAR) in Lake Rauchuagytgyn. Pearson correlation coefficients indicated by r in black when correlation was significant (p<0.05), and in red italic when p>0.05 (insignificant).**


**Table 1: Radiocarbon dates from sediment core EN18218 from Vyse et al. (2021) used to generate age-depth relationships and sedimentation rates in LANDO.**

| Lab code | Sample ID | Composite depth (cm) | Radiocarbon age with error ($^{14}$C years BP) | Sample type |
|----------|-----------|---------------------|---------------------|-------------|
| AWI - 5627.1.1 | EN18218-1 Surface 0-0.5 cm | 0.25 | 785 ± 31 | Bulk, TOC |
| AWI - 2998.1.1 | EN18218-2_0-100_20-20.5 | 18.75 | 2787 ± 33 | Bulk, TOC |
| AWI - 2999.1.1 | EN18218-2_0-100_36.5-37 | 35.25 | 3629 ± 33 | Bulk, TOC |
| AWI - 3000.1.1 | EN18218-2_0-100_61-61.5 | 59.75 | 3832 ± 33 | Bulk, TOC |
| AWI - 3003.1.1 | EN18218-2_100-200_140-140.5 | 138.75 | 5074 ± 34 | Bulk, TOC |
| AWI - 3004.1.1 | EN18218-2_100-200_164-164.5 | 162.75 | 5382 ± 34 | Bulk, TOC |
| AWI - 3005.1.1 | EN18218-2_100-200_189-189.5 | 187.75 | 5852 ± 34 | Bulk, TOC |
| AWI - 3006.1.1 | EN18218-2_200-240_222.5-223 | 221.75 | 6472 ± 35 | Bulk, TOC |
| AWI - 3007.1.2 | EN18218-3_0-100_15-15.5 | 248.75 | 8872 ± 37 | Bulk, TOC |
| AWI - 3008.1.1 | EN18218-3_0-100_37-37.5 | 270.75 | 9085 ± 37 | Bulk, TOC |
| AWI - 3009.1.1 | EN18218-3_0-100_59.5-60 | 293.25 | 9516 ± 38 | Bulk, TOC |
| AWI - 3010.1.1 | EN18218-3_0-100_83-83.5 | 316.75 | 9901 ± 39 | Bulk, TOC |
| AWI - 3011.1.1 | EN18218-3_100-200_105-105.5 | 338.75 | 10197 ± 39 | Bulk, TOC |
| AWI - 3012.11 | EN18218-3_100-200_129-129.5 | 362.75 | 11687 ± 30 | Bulk, TOC |
| AWI - 3013.1.1 | EN18218-3_100-200_150-150.5 | 383.75 | 12205 ± 46 | Bulk, TOC |
| AWI - 3014.1.1 | EN18218-3_100-200_171-171.5 | 404.75 | 13017 ± 48 | Bulk, TOC |
| AWI - 3015.1.1 | EN18218-3_200-292_210-210.5 | 443.75 | 14330 ± 52 | Bulk, TOC |
| AWI - 3016.1.1 | EN18218-3_200-292_239-239.5 | 474.75 | 15686 ± 48 | Bulk, TOC |





| AWI - 3017.1.1 | EN18218-3_200-292_270-270.5 | 503.75 | 17708 ± 56 | Bulk, TOC |
|---|---|---|---|---|
| AWI - 3018.1.1 | EN18218-4_0-100_35-35.5 | 536.25 | 18000 ± 55 | Bulk, TOC |
| AWI - 3019.1.1 | EN18218-4_0-100_64.5-65 | 565.75 | 22649 ± 66 | Bulk, TOC |
| AWI - 3020.1.1 | EN18218-4_0-100_95-95.5 | 596.25 | 21786 ± 204 | Bulk, TOC |
| AWI - 3021.1.1 | EN18218-4_100-163_123-123.5 | 624.25 | 25689 ± 325 | Bulk, TOC |
| AWI - 3022.1.1 | EN18218-4_100-163_145-145.5 | 646.25 | 25081 ± 300 | Bulk, TOC |

**Table 2:  Fallout radionuclide concentrations in the short core 16-KP-04-L19B**

| Depth | | $^{210}$Pb | | | | | | $^{137}$Cs | |
|---|---|---|---|---|---|---|---|---|---|
| | | Total | | Unsupported | | Supported | | | |
| cm | g cm$^{-2}$ | Bq kg$^{-1}$ | ± | Bq kg$^{-1}$ | ± | Bq kg$^{-1}$ | ± | Bq kg$^{-1}$ | ± |
| 1.25 | 0.29 | 285.3 | 13.8 | 222.5 | 14.1 | 62.7 | 2.6 | 47.3 | 2.3 |
| 1.75 | 0.40 | 221.8 | 10.0 | 159.0 | 10.4 | 62.8 | 2.8 | 54.2 | 1.8 |
| 2.25 | 0.52 | 240.5 | 13.4 | 176.6 | 13.6 | 63.8 | 2.8 | 60.0 | 2.6 |
| 2.75 | 0.62 | 247.6 | 12.1 | 183.1 | 12.4 | 64.4 | 2.8 | 70.1 | 2.6 |
| 3.25 | 0.75 | 138.9 | 10.6 | 73.8 | 10.8 | 65.1 | 2.4 | 79.5 | 2.4 |
| 3.75 | 0.94 | 106.0 | 6.8 | 39.7 | 7.1 | 66.3 | 2.2 | 52.1 | 1.4 |
| 4.25 | 1.15 | 103.6 | 8.9 | 40.1 | 9.1 | 63.5 | 1.9 | 23.8 | 1.6 |
| 4.75 | 1.34 | 113.1 | 6.8 | 52.3 | 7.1 | 60.8 | 1.9 | 15.6 | 1.2 |
| 5.25 | 1.54 | 83.0 | 7.2 | 25.3 | 7.4 | 57.7 | 1.8 | 9.4 | 0.9 |
| 5.75 | 1.75 | 69.9 | 6.7 | 11.7 | 6.9 | 58.2 | 1.8 | 2.1 | 0.8 |
| 6.25 | 1.93 | 82.5 | 6.4 | 23.4 | 6.7 | 59.1 | 1.9 | 2.2 | 0.9 |
| 6.75 | 2.11 | 68.1 | 7.7 | 8.7 | 7.9 | 59.4 | 1.8 | 2.0 | 1.1 |
| 7.25 | 2.31 | 67.6 | 5.6 | 6.3 | 5.9 | 61.3 | 1.9 | 1.2 | 0.7 |
| 7.75 | 2.51 | 66.0 | 7.7 | 3.7 | 7.9 | 62.3 | 1.7 | 0.0 | 0.0 |
| 8.25 | 2.71 | 62.3 | 6.0 | -0.2 | 6.2 | 62.4 | 1.6 | 0.8 | 0.9 |
| 8.75 | 2.91 | 58.2 | 4.0 | -1.9 | 4.3 | 60.0 | 1.6 | 0.2 | 0.7 |
| 9.25 | 3.12 | 70.3 | 7.3 | 11.5 | 7.5 | 58.8 | 1.7 | 1.5 | 0.8 |
| 9.75 | 3.32 | 55.5 | 6.5 | -4.0 | 6.8 | 59.5 | 1.9 | 0.0 | 0.0 |
| 10.25 | 3.53 | 64.2 | 6.1 | 4.6 | 6.3 | 59.7 | 1.8 | 0.0 | 0.0 |