# Peer review of "Diatom responses and geochemical feedbacks to environmental changes at Lake Rauchuagytgyn (Far East Russian Arctic)"

_EGUsphere, 2022_

## Author Response (AR1)

**Author revision notes to *Biskaborn et al.: Diatom responses and geochemical feedbacks to environmental changes at Lake Rauchuagytgyn (Far East Russian Arctic)**

**Associate editor decision: Reconsider after major revisions**
**by Petr Kuneš**

Thank you for your thorough responses to both reviews. I am still not entirely happy with how you handled the zonation of diatom assemblages. Perhaps the Bray-Curtis dissimilarity is not well fit for your case after all. Have you tried one of the more classical distances, such as Hellinger, or even the application of log transformation on Euclidean distance? It makes little sense to me to apply Bray-Curtis, get an indefinite number of significant zones and then to abandon that analysis. In fact, you use squared-root transformation to perform your PCA!
And I am also wondering since one reviewer suggested the combination of short and long cores, should not these be zoned together?
Please prepare the major revision of your manuscript, including addressing my comments.

> **Dear Petr Kuneš**
>
> Thank you very much for allowing us to resubmit the manuscript. We carefully addressed all issues and implemented the suggested changes in the revised version of the manuscript. Please find our detailed point-to-point answers to the referee comments below. Referee comments in black, our answers right-indented in blue.
>
> Thank you for your additional comment on the zonation of diatom assemblages. We agree that bray-curtis might not be the optimal choice in this case. We tested all suggestions you had and found that Euclidean distance together with log-transformation revealed the most reasonable results, in fact it reduced the total number of diatom zones in the manuscript to 8 (before 10) and fixed also a bad zone we ignored before because it was based on only one sample. We found log transformation more useful than square-root for CONISS because it down-weights the abundant species more strongly. In the PCA we didn't want to overdue downweighing because we want to extract the main signal of important species, so we stick to square-root transformation in this analysis. Thank you again for your comment. In fact this helped us to find the same procedure we applied already in a recently published paper (Biskaborn et al., 2021).
>
> We changed the method text accordingly:
> "To create diatom zones along the cores we used the package 'rioja' for constrained incremental sums-of-squares clustering (CONISS) based on Euclidean dissimilarity after log transformation of species percentage data to downweigh abundant species (Grimm, 1987). Attribution of diatom zones along the core depth was guided by CONISS results, while the total number of zones in the cores is referring to

meaningful chronologies in the region (Andreev et al., 2021; Anderson and Lozhkin, 2015; Andreev et al., 2012).
We used the 'decorana' function in the package 'vegan' (Oksanen et al., 2020) to apply a detrended correspondence analysis (DCA) on percentage data and calculated gradient length in standard deviation units (SD EN18218 = 2.36; SD 16-KP-04-L19B = 1.25). According to the threshold suggested by Birks (2010) we chose principal component analysis (PCA) to reveal major trends in the data. In the PCA we also chose Euclidean distance, but square-root transformation of the data to downweigh abundant species less aggressively than log transformation performed in CONISS."

We also changed the zonation in the figure, and everywhere else in the text. We had to modify some a few paragraphs in the discussion, due to the reduction of the two earliest zones A and B to " 1 " and H, I, and J to " 7 " and " 8 ". The main interpretation in these zones was not changed, but fine-sanded according to the other comments of the Reviewers.

We understand your suggestion related to the two data sets and combination of figures. However, in the case of Rauchuagytgyn, a combination of long and short core records does not make entirely sense, because the cores come from different locations 220 m away from each other and the features of the basin and sediment at these spots are different (Vyse et al., 2021; Biskaborn et al., 2019). It would imply that there is a change in the time series, which is rather a spatial variability signal than a true shift over time. We therefore cannot combine the data sets for CONISS zonation. In the new version we named the zones 1-6 in the long core, and 7-8 in the short core, because they are following up in one chronological order (but at different sites). We made it more clear in the discussion: *"… the short core 16-KP-04-L19B about 220 m E of the long core position. The surface sediments in this area of the lake were slightly different but also dominated by the same Lindavia and Aulacoseira species …"*

We performed one more English proof reading of the text. All changes in the manuscript are highlighted in yellow. We also provided a list with updated data sets available on PANGAEA in the "data availability" section in the manuscript.

*"Datasets used in this study are accessible on PANGAEA.*

*Long core EN18218*
*Diatoms: https://doi.pangaea.de/10.1594/PANGAEA.953126*
*Nitrogen: https://doi.pangaea.de/10.1594/PANGAEA.953129*
*Mercury: https://doi.pangaea.de/10.1594/PANGAEA.953130*
*Dating and accumulation rates: https://doi.pangaea.de/10.1594/PANGAEA.953132*
*Biogeochemical data from Vyse et al. (2021):*
*https://doi.pangaea.de/10.1594/PANGAEA.929719*

*Short core 16-KP-04-L19B*
*Nitrogen, carbon, mercury: https://doi.pangaea.de/10.1594/PANGAEA.953134*
*Diatoms: https://doi.pangaea.de/10.1594/PANGAEA.953138*

*Lead-210 and Caesium-137 data: https://doi.pangaea.de/10.1594/PANGAEA.953139*
*Dating and accumulation rates: https://doi.pangaea.de/10.1594/PANGAEA.953142"*

Following reviewer recommendations, we shifted the correlation figure and also
provided hydrochemical parameters to supplementary material.

With kind regards, on behalf of all authors,
Boris Biskaborn

**Referee #1**
**General comments to the authors:**

This is an interesting and data-rich research paper from a remote region of Chukotka, far
eastern Russian Arctic. Here, the environmental history of Lake Rauchuagytgyn was
reconstructed during the last 29 ka years, using 14C-dated sediment records of diatoms
frustules together with organic carbon, nitrogen and mercury accumulation rates. Diatom
assemblages responded to the major Pleistocene and Holocene environmental changes
through shifts in species composition and abundance. It appears that organic carbon in lake
sediments has largely autochthonous origin, as it is strongly correlated with the diatom
frustule accumulation rate. This implies that this type of arctic lake may play an important
role as a carbon sink as most organic carbon (OC) is effectively buried in the sediments.
However, OC concentrations in the sediments do not reflect the amount of carbon dioxide
emitted during the ice melt, and this may be substantial.
I suggest that you change your discussion, conclusions and the abstract reflecting on existing
uncertainty regarding the role of arctic lakes in the global carbon cycle.
The age-depth model of the long core, which was based on 23 14C bulk organic carbon
dates in a 651.75 cm core, was developed using LANDO modelling. This is in a broad
agreement with the earlier published core chronology using different modelling and this
gives extra confidence in the results. The short core was dated using 210Pb and 137Cs. The
short core chronology is uncertain for the 7-11 cm interval.
Overall, this is a valuable and thorough contribution to the series of recent publications
from this remote arctic region, which may play an important role in the global climate
change. It deserves a publication. Substantial uncertainty remains, however, regarding the
fate of carbon in artic lakes and their role as sinks or emitters of carbon.

**Author final response**
**Dear Reviewer #1**
Thank you very much for reviewing the manuscript and your work invested in giving
feedback to our study. We are specifically grateful for the comment on remaining
uncertainties on the fate $CO_2$ that is not only accumulated in the sediment but also
released after ice-brake up of the lake. We understand that we used misleading
wording when using "during warming", but we actually were addressing millennial
scale Holocene warming. We agree that we did not discuss this aspect enough in the
manuscript, even though we considered it in the overall study, mainly based on
results of the long core record spanning multiple millennia. We now made sure that
our findings point to long-term trends, i.e. storage of carbon over millennia in the

Holocene interglacial part of the core that is well correlated with diatom valve production and thus primary production.

Accordingly, we included explanations in the abstract:

*"We conclude that, if increased short-term emissions are neglected, pristine Arctic lake systems can potentially serve as long-term $CO_2$ and Hg sinks during warm climate episodes driven by insolation-enhanced within-lake primary productivity."*

In the introduction:

*"There is an ongoing discussion about the role of Arctic lakes for the carbon cycle. Thermokarst basins are believed to have switched from a net source to a sink during the mid-Holocene ca. 5000 years ago related to permafrost dynamics (Anthony et al., 2014). Glacial lakes are often larger and well oxygenized and thus are considered to strongly contribute to the modern CO2 emission in the Arctic landscape (Tan et al., 2017; Wik et al., 2016). Differences in drivers of bioproductivity, e.g. land-use in Europe (Vihma et al., 2016), accumulation rate and preservation of sedimentary carbon, e.g. during the ice-melt (Spangenberg et al., 2021), still lead to a high sink-source variability across temporal scales. To help gain insights into the fate of carbon accumulated in northern lakes, we provide a high-resolution study of a sediment core from Lake Rauchuagytgyn."*

And conclusion:

*"From our study we infer that bulk carbon accumulation is represented by climate-enhanced within-lake primary productivity. Thus, pristine boreal lake systems potentially can serve as long-term $CO_2$ sinks if short-term fluctuations are disregarded. Lake basins also represent disposal sites for heavy metal contaminants."*

We think that our edits will sufficiently address the uncertainty related to the modern role of Arctic lakes, especially the methane and CO2 release. Nevertheless, the fact that there is a larger carbon pool preserved - existing - within Holocene sediments (0.15 Mt) than compared to Pleistocene sediments (0.1 Mt), is itself an 'indisputable' evidence of carbon removal during warm episodes, seen at long time scales of course.

Referee #1
Detailed comments to the authors:

The manuscript is well written for the most part, although some of the sentences need clarifying or re-writing. I highlighted those in the attached pdf copy. In addition, there are several minor grammar mistakes which I corrected and highlighted in the attached pdf file. Below I outlined sentences which need re-writing and other comments.

I suggest combining short and long core diatom and biochemistry data to facilitate visual interpretation of the results, please see the details below. This will also reduce the number of figures in this manuscript, which is quite high. My other concern is the way sedimentation rates were calculated. It is not well described in Methods. I give the details below.

Thank you for your help in improving our manuscript. We carefully used your detailed comments to revise the manuscript and provided point-to-point answers

and explanations of what we change, listed below. Yes, we now combined long core and short core graphs to reduce the total number of figures, and we described better our calculations using equations.

Abstract:
Line 37. Please change this sentence as uncertainty exists regarding the role of arcitic lakes in the global carbon cycle.

Thank you for the comment – we changed it to "We conclude that, if increased short-term emissions are neglected, pristine Arctic lake systems can potentially serve as long-term CO2 and Hg sinks during warming climate driven by insolation-enhanced within-lake primary productivity".

Methods.
Lines 195-205.
I suggest replacing the text with formulae and short justification of your calculations of sedimentation rates. You need to clearly describe how sedimentation rates were calculated to ensure the validity of the results.

Thank you for the comment. We agree and added the equations used to calculate sedimentation rates SR and mass accumulation rates MAR, and also described all terms used in these equations so that all accumulation rate values of diatoms, carbon, and mercury are reproducible in a mathematical way:

To calculate accumulation rates, we first computed dry mass accumulation rates (MAR, in $g\ cm^{-2}\ a^{-1}$) using equation 1.

$$MAR = DBD \times SR \qquad (1)$$

where DBD is dry bulk density (in $g\ cm^{-3}$) and SR is sedimentation rate (in $cm\ a^{-1}$). We derived SR from age-depth modelling in a standard procedure according to equation 2 (Pfalz et al., 2022).

$$SR\ (x_i) = \frac{depth(x_i) - depth\ (x_{i-1})}{age(x_i) - age(x_{i-1})} \qquad (2)$$

The value $x_i$ represents the layer of interest within a sediment core for which the SR calculation is necessary, while $x_{i-1}$ is the previous layers.

Results.
Please combine Figures 5 and 6 into one indicating a gap between two cores. This would facilitate better visualization of the floristic changes.
Similarly, please combine Figures 8 and 9 into one figure, this would ease interpretation of the results.
Line 284. You refer here to Figure 5, not Figure 9.

Thank you for the suggestion to combine the figures associated to the short core and the long core. These cores are from slightly different area of the lake, that explains changes in the floristic mode especially taken into account the high morphological variability of the lake floor. However, we combined the diatom assemblage graphs

and the biochemistry plots in a way that the short core is above the long core. This will help the reader to more easily identify the chronology using the y-axes.
We also adopted all Figure numbers to the new (lower) total number of figures used in the manuscript.

Discussion.
Section 5.2.
Consider reviewing this section, it requires clarifying and re-writing. Use combined Figure made from Figures 8 and 9 to describe the changes and correlations between the profiles.
Thank you very much for pointing this out. Yes, we substantially reviewed and edited this entire section. Your detailed comments helped us very much.

Lines 400-405. You need to re-write this passage about sediment accumulation rates, it is not entirely clear.
Thank you very much. We changed this paragraph to: "Accumulation rates (AR) in sediment basins are generally uncertain due to limitations in precise age-model interpolations (Sadler, 1981). In addition, diatom concentrations are expressed as numbers of frustules (Battarbee et al., 2001) regardless of the weight and volume of the shells. Accordingly, one cannot directly infer biomass from count-based valve accumulation, as valves vary considerably in size among and within species (Birks, 2010)".

Line 407. This sentence about lake ontogeny (which is lake development) looks incomplete to me. You need to explain how lake ontogeny changes, this is a process.
Thank you. We changed this sentence to: "We showed above that the Rauchuagytgyn sedimentary record shows a tendency toward successional lake development in response to long-term changes of the landscape and ecosystem adaptation (Brenner and Escobar, 2009). Therefore, unknown deviations in the linkage between the mass of carbon stored and number of diatom valves observed are likely to appear".

Lines 409-410 – Similarly, it is difficult to understand these sentences. You need to think clearly what you are trying to say here, and re-write this passage. I highlighted it in the pdf copy.
Thank you for pointing this out. We deleted this sentence.

Lines 462-465. These two sentences require reviewing and rewriting, it is not entirely clear what is the meaning there.
Thank you very much. We agree and deleted one of these sentences and reworded the other to: "Our study revealed a positive feedback mechanism between long-term climate amelioration and diatom-driven sink of organic matter."

Figures.
Figure 10 is not easy to interpret, I do not think that this Figure is necessary, I suggest removing it.
It would be better to combine Figures 8 and 9. The combined Figure can be used to describe and discuss the trends in the profile changes and correlations between them.

If you wish to display the correlations, you can use a table.

> Thank you for your suggestion. We now combined Figures 8 and 9 as suggested. We removed the correlation plots to the supplement material because we believe it contains important information highlighting the good correlation during Holocene warm and missing correlation in the Pleistocene cold episode.

Please also note the supplement to this comment:
https://egusphere.copernicus.org/preprints/2022/egusphere-2022-985/egusphere-2022-985-RC1-supplement.pdf

> Thank you for your additional comments and corrections provided in the PDF. We saw all of them and adopted the corrections. Thank you also for the comment on how we could try to approach biomass by diatom measurements for follow-up analyses. It would definitely be interesting to measure sizes of dominant species and will hopefully be possible to do with increasing use of high-res microscopes and data-science methods. Same for measuring pigments.

**Author revision notes to *Biskaborn et al.: Diatom responses and geochemical feedbacks to environmental changes at Lake Rauchuagytgyn (Far East Russian Arctic)***

**Referee #2**

**General comments to the Author**

This manuscript explores past environmental changes during the last 29k years based on diatom and geochemical records of a well-dated lake sediment core in Chukotka, which is a less investigated region. The results are interesting, particularly the relationships between diatom accumulation rates, organic carbon accumulation rates and mercury accumulation rates. I have one major concern on the driving force on diatom flora shifts (see details as follows). I suggest that this manuscript can be acceptable after major revions.

Major concern:

Although authors have provided potential driving forces for diatom flora shift, the mechanism is still ambiguous. The major trends in diatom flora are that an increase in benthic species during the early Holocene, and then the gradual replacement of benthic taxa by planktonic species. This general trend might be closely linked to temperature- driven changes in ice-cover period. For example, short ice-cover period in the early Holocene promote light penetration and the availability of littoral habitats.

In addition, effects of DOC on diatom flora should be considered during lake ontogeny since the deglaciation. DOC can be an important resilience against from external driving forces.

Some references might be useful

Engstrom, D. R., et al. (2000). "Chemical and biological trends during lake evolution in recently deglaciated terrain." Nature 408(6809): 161-166.

Hu, Z., et al. (2018). "The Landscape–Atmosphere Continuum Determines Ecological Change in Alpine Lakes of SE Tibet." Ecosystems 21(5): 839-851.

Chen, X., et al. (2018). "Direct and indirect effects of Holocene climate variations on catchment and lake processes of a treeline lake, SW China." Palaeogeography, Palaeoclimatology, Palaeoecology 502: 119-129.

Wischnewski, J., et al. (2011). "Terrestrial and aquatic responses to climate change and human impact on the southeastern Tibetan Plateau during the past two centuries." Global Change Biology 17(11): 3376-3391.

I suggest that authors could clarify the major underlying driving forces of diatom flora shifts, specify the meanings of PC1 and PC2.

> **Author final response**
> **Dear Reviewer #2**
> Thank you very much for taking your time to review the manuscript. We are grateful for the comments you provided to ecology and environmental interpretation of diatom community shifts. We find that your comments on the increase of benthic species in course of changing ice cover period in the Early Holocene fits well into our discussion that was focused on stratification. We extended our discussion taking into account the ice-cover related mechanisms as well as DOC as a factor driving diatom species changes. We also included the literature you recommended while doing so. We added: *"Over the deglaciation period, in parallel to development of catchment vegetation, the lake ontogeny was likely driven by changes in the load of dissolved*

*organic carbon (DOC). As shown in lake evolution studies (Engstrom et al., 2000), young lakes in freshly deglaciated terrain have low DOC and rather alkaline conditions, which is reflected by the benthic species assemblage in the record, such as fragilariod species successively accompanied by Encyonopsis descriptiformis and Brachysira neoexilis."*

We also used the PCA, i.e. PC1, to explain the major change at the Pleistocene-Holocene boundary (from planktonic to benthic taxa). The revised discussion of underlying driving forces of diatom shifts is now in accord with the PCA results as suggested: *"The P/H is also characterized by a distinct increase of the first axis sample scores of the PCA (Fig. 8) pointing to the most prominent increase benthic diatom taxa in the record. The PCA biplot depicts grouping of planktonic Lindavia versus benthic Staurosira and Psammothidium species along the primary axis while Aulacoseira species are oriented along the secondary axis (Fig. 7). This general shift to benthic communities can be explained by temperature-driven changes in the duration of the ice-cover period. Longer open-water seasons in the Early Holocene promote light penetration and the availability of littoral habitats, while input of DOC and nutrients enhances benthic production in the littoral zone (Hu et al., 2018; Engstrom et al., 2000)."*

And in the last sentence of the discussion: *"This is amplified by the fact that boreal lakes have either already passed important ecosystem thresholds, or are about to exceed ecological tipping points upon further warming (Wischnewski et al., 2011) and are believed to not represent pristine ecosystems anymore (Smol et al., 2005)."*

Referee #2
Other minor revisions:

1) L31-32: the responses to climate events are not very clear, probably due to low resolution of diatom records

Thank you for the comment. We agree and changed to "moderate responses". We think that this wording reflects the signals in the diatom record, fitting well to the age model and what is known from climate history in the region.

2) L33-34: 'human-induced environmental change', please specify, atmospheric deposition?

Thank you for the comment. We agree, better to make this clear, we changed the sentence to: "The short core data likely suggest recent change of the diatom community at the beginning of the 20th Century related to human-induced warming but only little evidence of atmospheric deposition of contaminants"

3) L34-35:C/N ratios are generally larger than 10 during the Holocene, suggestive of the mixture of within lake production and terrestrial organic matter. Therefore, it should be cautious to draw this conclusion.

Thank you. We visited this lake and know that there is very little terrestrial catchment vegetation. However, you are right that the data alone cannot prove it. We therefore reworded to be more precise in this statement: *"Significant correlation between DAR and OCAR in the Holocene interglacial indicates within-lake*

*bioproduction represents bulk organic carbon deposited in the lake sediment."* Furthermore we checked the related statement in the conclusion and find that it is still correct as is.

4) Section 2 Study site: Please provide more details on aquatic plants in this lake. Are there macrophytes or mosses around the lake shore? This is important to explain the development of benthic diatoms during the Holocene. In addition, water chemistry data should be provided, such as pH, conductivity, and dissolved organic carbon,

Thank you for the statement. Unfortunately, we have no reliable information on the water plants from our coring expedition and cannot visit this lake again, as it is located in the Far East Russian Arctic. However, we have the water chemistry data that you requested and provided it in the supplementary material to the paper. We also added the main hydrochemical preferences of the lake in the study site chapter: *"Hydrochemical data from July 2018 (supplementary material S2) showed that the lake water had dilute freshwater with low conductivity (85.5 μs cm$^{-1}$), medium transparency (secchi depth 3.9 m), slightly alkaline conditions (pH 7.8), and low dissolved organic carbon (0.9 mg L$^{-1}$)"*.
We also used the values in the discussion of the core record:" *As shown in lake evolution studies (Engstrom et al., 2000), young lakes in freshly deglaciated terrain have low DOC and rather alkaline conditions, which is reflected by the benthic species assemblage in the record, such as fragilariod species successively accompanied by Encyonopsis descriptiformis and Brachysira neoexilis. Modern DOC measured in July 2018 (0.9 mg L-1) clearly below the global lake average of 3.9 mg L−1 (Toming et al., 2020) together with other hydrochemical parameters (supplementary table S2) indicates an overall dilute and alkaline lake system, suggesting even depleted conditions in the past."*.

5) Section 3.5 Data processing and statistics: L 173, generally, square-root transformation of percentage data were used in CONISS, please check. In addition, the number of zones should be tested by the broken stick model (Bennett, 1996).
Bennett, K. D. (1996). "Determination of the number of zones in a biostratigraphical sequence." New Phytologist 132(1): 155-170.

Thank you for the comment. We agree and re-performed the CONISS analyses for both cores based on percentage data. We also changed bray-curtis to Euclidean distance as suggested by Petr Kuneš (please also have a look to our reply to the Editor). The results are slightly different and we reduced the number of zones from 10 to 8.

Following your advice, we tested the broken stick statistic for the long core

[Figure]

And also applied it to the short core

[Figure]

However, we doubt that – in this case – the broken stick analysis provides valuable information on the number of useful CONISS zones. We assume that the hierarchical clustering and stratigraphical constrained order of zones causes complications for this statistical method. We describe that we use the CONISS clustering as a "guide" to establish the zones. But the final number of zones (or possible sub-zones) is given by the entire 'holistic' view on the data and the known chronological variability in the region, i.e. is done by us as the authors. The precise selection of boundary is still following the revised CONISS analysis, i.e. some previous zones were cancelled. We are grateful for your comment that surely made the zonation more solid (using the percentage data and checking for the broken stick behaviour in the data set). We modified the text in the method section accordingly: "*To create diatom zones along the cores we used the package 'rioja' for constrained incremental sums-of-squares clustering (CONISS) based on Euclidean dissimilarity after log transformation of species percentage data to downweigh abundant species (Grimm, 1987). Attribution of diatom zones along the core depth was guided by CONISS results, while the total number of zones in the cores is referring to meaningful chronologies in the region (Andreev et al., 2021; Anderson and Lozhkin, 2015; Andreev et al., 2012)."*

6) L243-246, please check the units of DVC and DAR, superscript should be used.

Thank you for finding this typo. We corrected it.

7) L330-331: thick ice due to long ice-cover period probably reduces light penetration?

Thank you. Yes, we agree that this also is a mechanism supporting low abundancy of benthic taxa. We added this to our interpretation: *"Low abundance of benthic diatoms may result from thick ice due to long ice-cover periods and reduced light penetration, as well as in-wash of clay during deglaciation (Vyse et al., 2021) leading to low-transparent and narrow littoral zones in an overall deep basin."*

8) L339-340: more detailed explanation for the linkage between diatom flora shift and climate. During this stage, the major change is the disappearance of Linvidavia bodanica and L. cyclopuncta

Thank you for the comment. We agree and added a more detailed explanation:*" Corresponding to the Younger Dryas (YD) period our diatom data show disappearance of L. bodanica and L. cyclopuncta but relative increase of heavy Aulacoseira valves (Fig. 5 and 8) indicating to turbulent water conditions. Complex diatom responses within the YD associated with increase of Aulacoseira species have been found in Lake Baikal (Mackay et al., 2022). In many boreal lakes YD cooling weakened lake thermal stratification leading to turbulent conditions resulting in similar diatom responses as observed in Lake Rauchuagytgyn (Neil and Lacourse, 2019)."*.

9) L354-355: rising alkalinity might be linked to enhanced chemical weathering intensity of bare rocks under warmer and wetter climate during the early and mid-Holocene

Thank you. We agree and modified this sentence to: *"Increased chemical weathering of bare rocks during warmer and wetter interglacial conditions, and the development of roots (Andreev et al., 2021) in fresh soils, all led to enhanced ion supply (Herzschuh et al., 2013) and eventually increased alkalinity of the lake water."*

10) L371: the influx of melted water during the spring and summer probably increase the mixing?

Thank you. We agree and modified this sentence to: *"Early ice-out, the influx of melt water during spring and summer associated to increased and longer spring circulation supported Aulacoseira species (Horn et al., 2011) and led to a distinct change in the Rauchuagytgyn species assemblage."*.

11) L396: Here, potential effects of nitrogen deposition?

Thank you. Yes, we added a sentence in this paragraph: *"As a pennate planktonic diatom, Tabellaria often responds with increased abundancy to atmospheric nitrogen deposition (Rühland et al., 2015), corresponding to increased nitrogen levels between 1970 and 1980 CE (Fig. 7a)."*

12) L488-489: prolonged ice-free period increases the availability of littoral habitats

Thank you. Yes, according to our modification of the discussion we changed this paragraph in the conclusion to:*" The Early Holocene diatom community reflects a shallower lake with larger littoral zones and higher alkalinity that we relate to prolonged ice-free periods and vegetation development in the catchment, supported by high carbon to nitrogen ratios."*

13) Figure 7: legends for the two figures are needed, depths can be changed to 'ages'

Thank you for the advice. We agree and added a legend that explains the symbols and colors used, i.e. attributing chronologies to the samples shown.

14) Figure 8: for the total percentage of light Lindavia, L. bodanica might be different from other species, since this taxon has relatively heavy valves, which are similar to some Aulacoseira species (see the review by Saros and Anderson, 2015, The ecology of the planktonic diatom Cyclotella and its implications for global environmental change studies).

Thank you for the hint. We knew about this before and this was why we did not include *L. bodanica* in the group of "light *Lindavia*", it was and is just represented as single species percentages.

References

Biskaborn, B. K., Nazarova, L., Kröger, T., Pestryakova, L. A., Syrykh, L., Pfalz, G., Herzschuh, U., and Diekmann, B.: Late Quaternary Climate Reconstruction and Lead-Lag Relationships of Biotic and Sediment-Geochemical Indicators at Lake Bolshoe Toko, Siberia, Frontiers in Earth Science, 9, 10.3389/feart.2021.737353, 2021.

Biskaborn, B. K., Nazarova, L., Pestryakova, L. A., Syrykh, L., Funck, K., Meyer, H., Chapligin, B., Vyse, S., Gorodnichev, R., Zakharov, E., Wang, R., Schwamborn, G., Bailey, H. L., and Diekmann, B.: Spatial distribution of environmental indicators in surface sediments of Lake Bolshoe Toko, Yakutia, Russia, Biogeosciences, 16, 4023-4049, 10.5194/bg-16-4023-2019, 2019.

Vyse, S. A., Herzschuh, U., Pfalz, G., Pestryakova, L. A., Diekmann, B., Nowaczyk, N., and Biskaborn, B. K.: Sediment and carbon accumulation in a glacial lake in Chukotka (Arctic Siberia) during the Late Pleistocene and Holocene: combining hydroacoustic profiling and down-core analyses, Biogeosciences, 18, 4791-4816, 10.5194/bg-18-4791-2021, 2021.

---

## Author Response (AR2)

**2nd Author revision notes to *Biskaborn et al.: Diatom responses and geochemical feedbacks to environmental changes at Lake Rauchuagytgyn (Far East Russian Arctic)***

**Associate editor decision: minor revision**

> **Author's response**
> **Dear Petr Kuneš**
>
> Thank you very much for the possibility to revise our manuscript a second time. We solved each of the issues brought up by Reviewer #2 and documented it in the point-to-point answer below. Reviewer's comments in black, our answers in blue and indented.
>
> With best regards
> Boris Biskaborn

**Report #1 Referee #2**

Authors have revised the manuscript based on early comments and suggestions. It can be acceptable after minor revisions.

> **Dear Reviewer #2**
>
> Thank you very much for reviewing the manuscript again and finding additional unclear points and typos. We are grateful for your volunteering support to increase the quality of our study presentation for Biogeosciences. We carefully addressed each of the issues you mentioned and documented it in the point-to-point answer below and a marked-up version of the manuscript.

Two moderate revisions

1) the explanation on the missing correlation between HgAR and DAR (L482-485) should be treated with caution. Before 16 ka BP, C/N ratios were relatively stable and less than 10, indicative the dominance of within-lake production in sediment organic matter.
Other potential reasons: improved preservation of nitrogen in prolonged ice cover period? alternatively, other algae (e.g., cold-tolerant chrysophytes or cyanobacteria) rather than diatoms flourished during the cold period

> Thank you very much for this additional comment. We deleted the previous sentence and changed this paragraph to: "*Mercury in tundra catchments is closely related to*

*non-vascular plants (Olson et al., 2019) and external supply of plant organic matter was reported to represented the main source of cold climate carbon deposition (Hughes-Allen et al., 2021). In Rauchuagytgyn, however, the higher amount of nitrogen detected in the pre-Holocene core section suggests one or both of the following two reasons: (1) within-lake aquatic production by algae other than well-preserved diatoms flourished during the glacial (Hernández-Almeida et al., 2015) and/or (2) the preservation of nitrogen was higher during the prolonged ice cover period (Kincaid et al., 2022), than during the interglacial."*

2) Keep diatom zone names consistent in Figure 5 and Figure 7.

Thank you very much for noticing this issue! Yes, we somehow missed to upload the updated version. We revised the figure carefully in detail including an update of the diatom zonation. We also updated the caption of this figure accordingly.

other minor revisions:

3) L32 add 'the' before Younger Dryas
   Thank you. We fixed this typo.

4) L123 add a dot after 'modelling approach'
   Thank you. We fixed this typo.

5) L294 italic type 'L. ocellata'
   Thank you. We fixed this typo.

6) L318 'highes' should be 'highest'
   Thank you. We fixed this typo.

7) L433 'inbetween' change to 'between'
   Thank you. We fixed this typo.